# EF-BV: A Unified Theory of Error Feedback and Variance Reduction Mechanisms for Biased and Unbiased Compression in Distributed Optimization

**Laurent Condat**[*]
KAUST

**Kai Yi**
KAUST

**Peter Richtárik**
King Abdullah University of Science and Technology (KAUST)
Thuwal 23955-6900, Kingdom of Saudi Arabia

## Abstract

In distributed or federated optimization and learning, communication between the different computing units is often the bottleneck and gradient compression is widely used to reduce the number of bits sent within each communication round of iterative methods. There are two classes of compression operators and separate algorithms making use of them. In the case of unbiased random compressors with bounded variance (e.g., rand-k), the DIANA algorithm of Mishchenko et al. (2019), which implements a variance reduction technique for handling the variance introduced by compression, is the current state of the art. In the case of biased and contractive compressors (e.g., top-k), the EF21 algorithm of Richtárik et al. (2021), which instead implements an error-feedback mechanism, is the current state of the art. These two classes of compression schemes and algorithms are distinct, with different analyses and proof techniques. In this paper, we unify them into a single framework and propose a new algorithm, recovering DIANA and EF21 as particular cases. Our general approach works with a new, larger class of compressors, which has two parameters, the bias and the variance, and includes unbiased and biased compressors as particular cases. This allows us to inherit the best of the two worlds: like EF21 and unlike DIANA, biased compressors, like top-k, whose good performance in practice is recognized, can be used. And like DIANA and unlike EF21, independent randomness at the compressors allows to mitigate the effects of compression, with the convergence rate improving when the number of parallel workers is large. This is the first time that an algorithm with all these features is proposed. We prove its linear convergence under certain conditions. Our approach takes a step towards better understanding of two so-far distinct worlds of communication-efficient distributed learning.

## 1 Introduction

In the big data era, the explosion in size and complexity of the data arises in parallel to a shift towards distributed computations (Verbraeken et al., 2021), as modern hardware increasingly relies on the power of uniting many parallel units into one system. For distributed optimization and learning tasks, specific issues arise, such as decentralized data storage. In the modern paradigm of *federated learning* (Konečný et al., 2016; McMahan et al., 2017; Kairouz et al., 2021; Li et al., 2020a), a potentially huge number of devices, with their owners' data stored on each of them, are involved in the

---

[*]Corresponding author. Contact: see https://lcondat.github.io/

36th Conference on Neural Information Processing Systems (NeurIPS 2022).

collaborative process of training a global machine learning model. The goal is to exploit the wealth of useful information lying in the *heterogeneous* data stored across the network of such devices. But users are increasingly sensitive to privacy concerns and prefer their data to never leave their devices. Thus, the devices have to *communicate* the right amount of information back and forth with a distant server, for this distributed learning process to work. Communication, which can be costly and slow, is the main bottleneck in this framework. So, it is of primary importance to devise novel algorithmic strategies, which are efficient in terms of computation and communication complexities. A natural and widely used idea is to make use of (lossy) *compression*, to reduce the size of the communicated messages (Alistarh et al., 2017; Wen et al., 2017; Wangni et al., 2018; Khaled & Richtárik, 2019; Albasyoni et al., 2020; Basu et al., 2020; Dutta et al., 2020; Sattler et al., 2020; Xu et al., 2021).

In this paper, we propose a stochastic gradient descent (SGD)-type method for distributed optimization, which uses possibly *biased* and randomized compression operators. Our algorithm is variance-reduced (Hanzely & Richtárik, 2019; Gorbunov et al., 2020a; Gower et al., 2020); that is, it converges to the exact solution, with fixed stepsizes, without any restrictive assumption on the functions to minimize.

**Problem.** We consider the convex optimization problem

$$\underset{x \in \mathbb{R}^d}{\text{minimize}} \ \underbrace{\frac{1}{n} \sum_{i=1}^{n} f_i(x)}_{f(x)} + R(x), \tag{1}$$

where $d \geq 1$ is the model dimension; $R : \mathbb{R}^d \to \mathbb{R} \cup \{+\infty\}$ is a proper, closed, convex function (Bauschke & Combettes, 2017), whose proximity operator $\text{prox}_{\gamma R} : x \mapsto \arg\min_{y \in \mathbb{R}^d} \left( \gamma R(y) + \frac{1}{2}\|x - y\|^2 \right)$ is easy to compute, for any $\gamma > 0$ (Parikh & Boyd, 2014; Condat et al., 2022a,b); $n \geq 1$ is the number of functions; each function $f_i : \mathbb{R}^d \to \mathbb{R}$ is convex and $L_i$-smooth, for some $L_i > 0$; that is, $f_i$ is differentiable on $\mathbb{R}^d$ and its gradient $\nabla f_i$ is $L_i$-Lipschitz continuous: for every $x \in \mathbb{R}^d$ and $x' \in \mathbb{R}^d$, $\|\nabla f_i(x) - \nabla f_i(x')\| \leq L_i \|x - x'\|$.

We set $L_{\max} := \max_i L_i$ and $\tilde{L} := \sqrt{\frac{1}{n} \sum_{i=1}^{n} L_i^2}$. The average function $f := \frac{1}{n} \sum_{i=1}^{n} f_i$ is $L$-smooth, for some $L \leq \tilde{L} \leq L_{\max}$. A minimizer of $f + R$ is supposed to exist. For any integer $m \geq 1$, we define the set $\mathcal{I}_m := \{1, \ldots, m\}$.

**Algorithms and Prior Work.** Distributed proximal SGD solves the problem (1) by iterating $x^{t+1} := \text{prox}_{\gamma R}\left(x^t - \frac{\gamma}{n} \sum_{i=1}^{n} g_i^t\right)$, where $\gamma$ is a stepsize and the vectors $g_i^t$ are possibly stochastic estimates of the gradients $\nabla f_i(x^t)$, which are cheap to compute or communicate. Compression is typically performed by the application of a possibly randomized operator $\mathcal{C} : \mathbb{R}^d \to \mathbb{R}^d$; that is, for any $x$, $\mathcal{C}(x)$ denotes a realization of a random variable, whose probability distribution depends on $x$. Compressors have the property that it is much easier/faster to transfer $\mathcal{C}(x)$ than the original message $x$. This can be achieved in several ways, for instance by sparsifying the input vector (Alistarh et al., 2018), or by quantizing its entries (Alistarh et al., 2017; Horváth et al., 2019; Gandikota et al., 2019; Mayekar & Tyagi, 2021; Saha et al., 2021), or via a combination of these and other approaches (Horváth et al., 2019; Albasyoni et al., 2020; Beznosikov et al., 2020). There are two classes of compression operators often studied in the literature: 1) unbiased compression operators, satisfying a variance bound proportional to the squared norm of the input vector, and 2) biased compression operators, whose square distortion is contractive with respect to the squared norm of the input vector; we present these two classes in Sections 2.1 and 2.2, respectively.

**Prior work: DIANA with unbiased compressors.** An important contribution to the field in the recent years is the variance-reduced SGD-type method called DIANA (Mishchenko et al., 2019), which uses unbiased compressors; it is shown in Fig. 1. DIANA was analyzed and extended in several ways, including bidirectional compression and acceleration, see, e.g., the work of Horváth et al. (2022); Mishchenko et al. (2020); Condat & Richtárik (2022); Philippenko & Dieuleveut (2020); Li et al. (2020b); Gorbunov et al. (2020b), and Gorbunov et al. (2020a); Khaled et al. (2020) for general theories about SGD-type methods, including variants using unbiased compression of (stochastic) gradients.

**Prior work: Error feedback with biased contractive compressors.** Our understanding of distributed optimization using biased compressors is more limited. The key complication comes from

Table 1: Desirable properties of a distributed compressed gradient descent algorithm converging to an exact solution of (1) and whether they are satisfied by the state-of-the-art algorithms DIANA and EF21 and their currently-known analysis, and the proposed algorithm EF-BV.

| | DIANA | EF21 | EF-BV |
|---|:---:|:---:|:---:|
| handles unbiased compressors in $\mathbb{U}(\omega)$ for any $\omega \geq 0$ | ✓ | ✓[1] | ✓ |
| handles biased contractive compressors in $\mathbb{B}(\alpha)$ for any $\alpha \in (0, 1]$ | ✗ | ✓ | ✓ |
| handles compressors in $\mathbb{C}(\eta, \omega)$ for any $\eta \in [0, 1), \omega \geq 0$ | ✗ | ✓[1] | ✓ |
| recovers DIANA and EF21 as particular cases | ✗ | ✗ | ✓ |
| the convergence rate improves when $n$ is large | ✓ | ✗ | ✓ |

[1]: with pre-scaling with $\lambda < 1$, so that $\mathcal{C}' = \lambda \mathcal{C} \in \mathbb{B}(\alpha)$ is used instead of $\mathcal{C}$

the fact that their naive use within methods like gradient descent can lead to divergence, as widely observed in practice, see also Example 1 of Beznosikov et al. (2020). *Error feedback* (EF), also called error compensation, techniques were proposed to fix this issue and obtain convergence, initially as heuristics (Seide et al., 2014). Theoretical advances have been made in the recent years in the analysis of EF, see the discussions and references in Richtárik et al. (2021) and Lin et al. (2022). But the question of whether it is possible to obtain a linearly convergent EF method in the general heterogeneous data setting, relying on biased compressors only, was still an open problem; until last year, 2021, when Richtárik et al. (2021) re-engineered the classical EF mechanism and came up with a new algorithm, called EF21. It was then extended in several ways, including by considering server-side compression, and the support of a regularizer $R$ in (1), by Fatkhullin et al. (2021). EF21 is shown in Fig. 1.

**Motivation and challenge.** While EF21 resolved an important theoretical problem in the field of distributed optimization with contractive compression, there are still several open questions. In particular, DIANA with independent random compressors has a $\frac{1}{n}$ factor in its iteration complexity; that is, it converges faster when the number $n$ of workers is larger. EF21 does not have this property: its convergence rate does not depend on $n$. Also, the convergence analysis and proof techniques for the two algorithms are different: the linear convergence analysis of DIANA relies on $\|x^t - x^\star\|^2$ and $\|h_i^t - \nabla f_i(x^\star)\|^2$ tending to zero, where $x^t$ is the estimate of the solution $x^\star$ at iteration $t$ and $h_i^t$ is the control variate maintained at node $i$, whereas the analysis of EF21 relies on $(f+R)(x^t)-(f+R)(x^\star)$ and $\|h_i^t - \nabla f_i(x^t)\|^2$ tending to zero, and under different assumptions. This work aims at filling this gap. That is, we want to address the following open problem:

*Is it possible to design an algorithm, which combines the advantages of DIANA and EF21? That is, such that:*

   a. *It deals with unbiased compressors, biased contractive compressors, and possibly even more.*

   b. *It recovers DIANA and EF21 as particular cases.*

   c. *Its convergence rate improves with $n$ large.*

**Contributions.** We answer positively this question and propose a new algorithm, which we name EF-BV, for *Error Feedback with Bias-Variance decomposition*, which for the first time satisfies the three aforementioned properties. This is illustrated in Tab. 1. More precisely, our contributions are:

1. We propose a new, larger class of compressors, which includes unbiased and biased contractive compressors as particular cases, and has two parameters, the **bias** $\eta$ and the **variance** $\omega$. A third parameter $\omega_{\mathrm{av}}$ describes the resulting variance from the parallel compressors after aggregation, and is key to getting faster convergence with large $n$, by allowing larger stepsizes than in EF21 in our framework.

2. We propose a new algorithm, named EF-BV, which exploits the properties of the compressors in the new class using two scaling parameters $\lambda$ and $\nu$. For particular values of $\lambda$ and $\nu$, EF21 and DIANA are recovered as particular cases. But by setting the values of $\lambda$ and $\nu$ optimally with respect to $\eta, \omega, \omega_{\mathrm{av}}$ in EF-BV, faster convergence can be obtained.

3. We prove linear convergence of EF-BV under a Kurdyka–Łojasiewicz condition of $f + R$, which is weaker than strong convexity of $f + R$. Even for EF21 and DIANA, this is new.

4. We provide new insights on EF21 and DIANA; for instance, we prove linear convergence of DIANA with biased compressors.

## 2 Compressors and their properties

### 2.1 Unbiased compressors

For every $\omega \geq 0$, we introduce the set $\mathbb{U}(\omega)$ of unbiased compressors, which are randomized operators of the form $\mathcal{C} : \mathbb{R}^d \to \mathbb{R}^d$, satisfying

$$\mathbb{E}[\mathcal{C}(x)] = x \quad \text{and} \quad \mathbb{E}[\|\mathcal{C}(x) - x\|^2] \leq \omega \|x\|^2, \quad \forall x \in \mathbb{R}^d, \tag{2}$$

where $\mathbb{E}[\cdot]$ denotes the expectation. The smaller $\omega$, the better, and $\omega = 0$ if and only if $\mathcal{C} = \mathrm{Id}$, the identity operator, which does not compress. We can remark that if $\mathcal{C} \in \mathbb{U}(\omega)$ is deterministic, then $\mathcal{C} = \mathrm{Id}$. So, unbiased compressors are random ones. A classical unbiased compressor is rand-$k$, for some $k \in \mathcal{I}_d$, which keeps $k$ elements chosen uniformly at random, multiplied by $\frac{d}{k}$, and sets the other elements to 0. It is easy to see that rand-$k$ belongs to $\mathbb{U}(\omega)$ with $\omega = \frac{d}{k} - 1$ (Beznosikov et al., 2020).

### 2.2 Biased contractive compressors

For every $\alpha \in (0, 1]$, we introduce the set $\mathbb{B}(\alpha)$ of biased contractive compressors, which are possibly randomized operators of the form $\mathcal{C} : \mathbb{R}^d \to \mathbb{R}^d$, satisfying

$$\mathbb{E}[\|\mathcal{C}(x) - x\|^2] \leq (1 - \alpha)\|x\|^2, \quad \forall x \in \mathbb{R}^d. \tag{3}$$

We use the term 'contractive' to reflect the fact that the squared norm in the left hand side of (3) is smaller, in expectation, than the one in the right hand side, since $1 - \alpha < 1$. This is not the case in (2), where $\omega$ can be arbitrarily large. The larger $\alpha$, the better, and $\alpha = 1$ if and only if $\mathcal{C} = \mathrm{Id}$. Biased compressors need not be random: a classical biased and deterministic compressor is top-$k$, for some $k \in \mathcal{I}_d$, which keeps the $k$ elements with largest absolute values unchanged and sets the other elements to 0. It is easy to see that top-$k$ belongs to $\mathbb{B}(\alpha)$ with $\alpha = \frac{k}{d}$ (Beznosikov et al., 2020).

### 2.3 New general class of compressors

We refer to Beznosikov et al. (2020), Table 1 in Safaryan et al. (2021), Zhang et al. (2021), Szlendak et al. (2022), for examples of compressors in $\mathbb{U}(\omega)$ or $\mathbb{B}(\alpha)$, and to Xu et al. (2020) for a system-oriented survey.

In this work, we introduce a new, more general class of compressors, ruled by 2 parameters, to allow for a finer characterization of their properties. Indeed, with any compressor $\mathcal{C}$, we can do a **bias-variance decomposition** of the compression error: for every $x \in \mathbb{R}^d$,

$$\mathbb{E}[\|\mathcal{C}(x) - x\|^2] = \underbrace{\|\mathbb{E}[\mathcal{C}(x)] - x\|^2}_{\text{bias}} + \underbrace{\mathbb{E}[\|\mathcal{C}(x) - \mathbb{E}[\mathcal{C}(x)]\|^2]}_{\text{variance}}. \tag{4}$$

Therefore, to better characterize the properties of compressors, we propose to parameterize these two parts, instead of only their sum: for every $\eta \in [0, 1)$ and $\omega \geq 0$, we introduce the new class $\mathbb{C}(\eta, \omega)$ of possibly random and biased operators, which are randomized operators of the form $\mathcal{C} : \mathbb{R}^d \to \mathbb{R}^d$, satisfying, for every $x \in \mathbb{R}^d$, the two properties:

(i) $\quad \|\mathbb{E}[\mathcal{C}(x)] - x\| \leq \eta \|x\|,$

(ii) $\quad \mathbb{E}[\|\mathcal{C}(x) - \mathbb{E}[\mathcal{C}(x)]\|^2] \leq \omega \|x\|^2.$

Thus, $\eta$ and $\omega$ control the relative bias and variance of the compressor, respectively. Note that $\omega$ can be arbitrarily large, but the compressors will be scaled in order to control the compression error, as we discuss in Sect. (2.5). On the other hand, we must have $\eta < 1$, since otherwise, no scaling can keep the compressor's discrepancy under control.

We have the following properties:

1. $\mathbb{C}(\eta, 0)$ is the class of deterministic compressors in $\mathbb{B}(\alpha)$, with $1 - \alpha = \eta^2$.

2. $\mathbb{C}(0, \omega) = \mathbb{U}(\omega)$, for every $\omega \geq 0$. In words, if its bias $\eta$ is zero, the compressor is unbiased with relative variance $\omega$.

3. Because of the bias-variance decomposition (4), if $\mathcal{C} \in \mathbb{C}(\eta, \omega)$ with $\eta^2 + \omega < 1$, then $\mathcal{C} \in \mathbb{B}(\alpha)$ with

$$1 - \alpha = \eta^2 + \omega. \qquad (5)$$

4. Conversely, if $\mathcal{C} \in \mathbb{B}(\alpha)$, one easily sees from (4) that there exist $\eta \leq \sqrt{1 - \alpha}$ and $\omega \leq 1 - \alpha$ such that $\mathcal{C} \in \mathbb{C}(\eta, \omega)$.

Thus, the new class $\mathbb{C}(\eta, \omega)$ generalizes the two previously known classes $\mathbb{U}(\omega)$ and $\mathbb{B}(\alpha)$. Actually, for compressors in $\mathbb{U}(\omega)$ and $\mathbb{B}(\alpha)$, we can just use DIANA and EF21, and our proposed algorithm EF-BV will stand out when the compressors are neither in $\mathbb{U}(\omega)$ nor in $\mathbb{B}(\alpha)$; that is why the strictly larger class $\mathbb{C}(\eta, \omega)$ is needed for our purpose.

We present new compressors in the class $\mathbb{C}(\eta, \omega)$ in Appendix A.

## 2.4 Average variance of several compressors

Given $n$ compressors $\mathcal{C}_i$, $i \in \mathcal{I}_n$, we are interested in how they behave in average. Indeed distributed algorithms consist, at every iteration, in compressing vectors in parallel, and then averaging them. Thus, we introduce the **average relative variance** $\omega_{\text{av}} \geq 0$ of the compressors, such that, for every $x_i \in \mathbb{R}^d$, $i \in \mathcal{I}_n$,

$$\mathbb{E}\left[\left\|\frac{1}{n}\sum_{i=1}^{n}\left(\mathcal{C}_i(x_i) - \mathbb{E}[\mathcal{C}_i(x_i)]\right)\right\|^2\right] \leq \frac{\omega_{\text{av}}}{n}\sum_{i=1}^{n}\|x_i\|^2. \qquad (6)$$

When every $\mathcal{C}_i$ is in $\mathbb{C}(\eta, \omega)$, for some $\eta \in [0, 1)$ and $\omega \geq 0$, then $\omega_{\text{av}} \leq \omega$; but $\omega_{\text{av}}$ can be much smaller than $\omega$, and we will exploit this property in EF-BV. We can also remark that $\frac{1}{n}\sum_{i=1}^{n}\mathcal{C}^i \in \mathbb{C}(\eta, \omega_{\text{av}})$.

An important property is the following: if the $\mathcal{C}_i$ are mutually independent, since the variance of a sum of random variables is the sum of their variances, then

$$\omega_{\text{av}} = \frac{\omega}{n}.$$

There are other cases where the compressors are dependent but $\omega_{\text{av}}$ is much smaller than $\omega$. Notably, the following setting can be used to model partial participation of $m$ among $n$ workers at every iteration of a distributed algorithm, for some $m \in \mathcal{I}_n$, with the $\mathcal{C}_i$ defined jointly as follows: for every $i \in \mathcal{I}_n$ and $x_i \in \mathbb{R}^d$,

$$\mathcal{C}_i(x_i) = \begin{cases} \frac{n}{m}x_i & \text{if } i \in \Omega \\ 0 & \text{otherwise} \end{cases},$$

where $\Omega$ is a subset of $\mathcal{I}_n$ of size $m$ chosen uniformly at random. This is sometimes called $m$-nice sampling (Richtárik & Takáč, 2016; Gower et al., 2021). Then every $\mathcal{C}_i$ belongs to $\mathbb{U}(\omega)$, with $\omega = \frac{n-m}{m}$, and, as shown for instance in Qian et al. (2019) and Proposition 1 in Condat & Richtárik (2022), (6) is satisfied with

$$\omega_{\text{av}} = \frac{n - m}{m(n - 1)} = \frac{\omega}{n - 1} \quad (= 0 \text{ if } n = m = 1).$$

## 2.5 Scaling compressors

A compressor $\mathcal{C} \in \mathbb{C}(\eta, \omega)$ does not necessarily belong to $\mathbb{B}(\alpha)$ for any $\alpha \in (0, 1]$, since $\omega$ can be arbitrarily large. Fortunately, the compression error can be kept under control by *scaling* the compressor; that is, using $\lambda\mathcal{C}$ instead of $\mathcal{C}$, for some scaling parameter $\lambda \leq 1$. We have:

**Proposition 1.** *Let $\mathcal{C} \in \mathbb{C}(\eta, \omega)$, for some $\eta \in [0, 1)$ and $\omega \geq 0$, and $\lambda \in (0, 1]$. Then $\lambda\mathcal{C} \in \mathbb{C}(\eta', \omega')$ with $\omega' = \lambda^2\omega$ and $\eta' = \lambda\eta + 1 - \lambda \in (0, 1]$.*

*Proof.* Let $x \in \mathbb{R}^d$. Then $\mathbb{E}\left[\|\lambda\mathcal{C}(x) - \mathbb{E}[\lambda\mathcal{C}(x)]\|^2\right] = \lambda^2\mathbb{E}\left[\|\mathcal{C}(x) - \mathbb{E}[\mathcal{C}(x)]\|^2\right] \leq \lambda^2\omega\|x\|^2$, and $\|\mathbb{E}[\lambda\mathcal{C}(x)] - x\| \leq \lambda\|\mathbb{E}[\mathcal{C}(x)] - x\| + (1 - \lambda)\|x\| \leq (\lambda\eta + 1 - \lambda)\|x\|$. $\qquad \square$

So, scaling deteriorates the bias, with $\eta' \geq \eta$, but linearly, whereas it reduces the variance $\omega$ quadratically. This is key, since the total error factor $(\eta')^2 + \omega'$ can be made smaller than 1 by choosing $\lambda$ sufficiently small:

**Proposition 2.** *Let $\mathcal{C} \in \mathbb{C}(\eta, \omega)$, for some $\eta \in [0, 1)$ and $\omega \geq 0$. There exists $\lambda \in (0, 1]$ such that $\lambda\mathcal{C} \in \mathbb{B}(\alpha)$, for some $\alpha = 1 - (1 - \lambda + \lambda\eta)^2 - \lambda^2\omega \in (0, 1]$, and the best such $\lambda$, maximizing $\alpha$, is*

$$\lambda^\star = \min\left(\frac{1 - \eta}{(1 - \eta)^2 + \omega}, 1\right).$$

*Proof.* We define the polynomial $P : \lambda \mapsto (1 - \lambda + \lambda\eta)^2 + \lambda^2\omega$. After Proposition 1 and the discussion in Sect. 2.3, we have to find $\lambda \in (0, 1]$ such that $P(\lambda) < 1$. Then $\lambda\mathcal{C} \in \mathbb{B}(\alpha)$, with $1 - \alpha = P(\lambda)$. Since $P$ is a strictly convex quadratic function on $[0, 1]$ with value 1 and negative derivative $\eta - 1$ at $\lambda = 0$, its minimum value on $[0, 1]$ is smaller than 1 and is attained at $\lambda^\star$, which either satisfies the first-order condition $0 = P'(\lambda) = -2(1 - \eta) + 2\lambda((1 - \eta)^2 + \omega)$, or, if this value is larger than 1, is equal to 1. $\square$

In particular, if $\eta = 0$, Proposition 2 recovers Lemma 8 of Richtárik et al. (2021), according to which, for $\mathcal{C} \in \mathbb{U}(\omega)$, $\lambda^\star\mathcal{C} \in \mathbb{B}(\frac{1}{\omega+1})$, with $\lambda^\star = \frac{1}{\omega+1}$. For instance, the scaled `rand-k` compressor, which keeps $k$ elements chosen uniformly at random unchanged and sets the other elements to 0, corresponds to scaling the unbiased `rand-k` compressor, seen in Sect. 2.1, by $\lambda = \frac{k}{d}$.

We can remark that scaling is used to mitigate the randomness of a compressor, but cannot be used to reduce its bias: if $\omega = 0$, $\lambda^\star = 1$.

Our new algorithm EF-BV will have two scaling parameters: $\lambda$, to mitigate the compression error in the control variates used for variance reduction, just like above, and $\nu$, to mitigate the error in the stochastic gradient estimate, in a similar way but with $\omega$ replaced by $\omega_{\mathrm{av}}$, since we have seen in Sect. 2.4 that $\omega_{\mathrm{av}}$ characterizes the randomness after averaging the outputs of several compressors.

## 3  Proposed algorithm EF-BV

We propose the algorithm EF-BV, shown in Fig. 1. It makes use of compressors $\mathcal{C}_i^t \in \mathbb{C}(\eta, \omega)$, for some $\eta \in [0, 1)$ and $\omega \geq 0$, and we introduce $\omega_{\mathrm{av}} \leq \omega$ such that (6) is satisfied. That is, for any $x \in \mathbb{R}^d$, the $\mathcal{C}_i^t(x)$, for $i \in \mathcal{I}_n$ and $t \geq 0$, are distinct random variables; their laws might be the same or not, but they all lie in the class $\mathbb{C}(\eta, \omega)$. Also, $\mathcal{C}_i^t(x)$ and $\mathcal{C}_{i'}^{t'}(x')$, for $t \neq t'$, are independent.

The compressors have the property that if their input is the zero vector, the compression error is zero, so we want to compress vectors that are close to zero, or at least converge to zero, to make the method variance-reduced. That is why each worker maintains a control variate $h_i^t$, converging, like $\nabla f_i(x^t)$, to $\nabla f_i(x^\star)$, for some solution $x^\star$. This way, the difference vectors $\nabla f_i(x^t) - h_i^t$ converge to zero, and these are the vectors that are going to be compressed. Thus, EF-BV takes the form of Distributed proximal SGD, with

$$g_i^t = h_i^t + \nu\mathcal{C}_i^t(\nabla f_i(x^t) - h_i^t),$$

where the scaling parameter $\nu$ will be used to make the compression error, averaged over $i$, small; that is, to make $g^{t+1} = \frac{1}{n}\sum_{i=1}^n g_i^t$ close to $\nabla f(x^t)$. In parallel, the control variates are updated similarly as

$$h_i^{t+1} = h_i^t + \lambda\mathcal{C}_i^t(\nabla f_i(x^t) - h_i^t),$$

where the scaling parameter $\lambda$ is used to make the compression error small, individually for each $i$; that is, to make $h_i^{t+1}$ close to $\nabla f_i(x^t)$.

### 3.1  EF21 as a particular case of EF-BV

There are two ways to recover EF21 as a particular case of EF-BV:

1. If the compressors $\mathcal{C}_i^t$ are in $\mathbb{B}(\alpha)$, for some $\alpha \in (0, 1]$, there is no need for scaling the compressors, and we can use EF-BV with $\lambda = \nu = 1$. Then the variable $h^t$ in EF-BV becomes redundant with the gradient estimate $g^t$ and we can only keep the latter, which yields EF21, as shown in Fig. 1.

| **Algorithm 1** EF-BV | **Algorithm 2** EF21 | **Algorithm 3** DIANA |
|---|---|---|
| proposed method | (Richtárik et al., 2021) | (Mishchenko et al., 2019) |

| | | |
|---|---|---|
| **Input:** $x^0, h_1^0, \ldots, h_n^0 \in \mathbb{R}^d$, $h^0 = \frac{1}{n}\sum_{i=1}^n h_i^0, \gamma > 0$, $\lambda \in (0,1], \nu \in (0,1]$ | **Input:** $x^0, h_1^0, \ldots, h_n^0 \in \mathbb{R}^d$, $h^0 = \frac{1}{n}\sum_{i=1}^n h_i^0, \gamma > 0$, | **Input:** $x^0, h_1^0, \ldots, h_n^0 \in \mathbb{R}^d$, $h^0 = \frac{1}{n}\sum_{i=1}^n h_i^0, \gamma > 0$, $\lambda \in (0,1]$ |
| **for** $t = 0, 1, \ldots$ **do** | **for** $t = 0, 1, \ldots$ **do** | **for** $t = 0, 1, \ldots$ **do** |
|   **for** $i = 1, 2, \ldots, n$ in parallel **do** |   **for** $i = 1, 2, \ldots, n$ in parallel **do** |   **for** $i = 1, 2, \ldots, n$ in parallel **do** |
|     $d_i^t := \mathcal{C}_i^t\big(\nabla f_i(x^t) - h_i^t\big)$ |     $d_i^t := \mathcal{C}_i^t\big(\nabla f_i(x^t) - h_i^t\big)$ |     $d_i^t := \mathcal{C}_i^t\big(\nabla f_i(x^t) - h_i^t\big)$ |
|     $h_i^{t+1} := h_i^t + \lambda d_i^t$ |     $h_i^{t+1} := h_i^t + d_i^t$ |     $h_i^{t+1} := h_i^t + \lambda d_i^t$ |
|     send $d_i^t$ to master |     send $d_i^t$ to master |     send $d_i^t$ to master |
|   **end for** |   **end for** |   **end for** |
|   at master: |   at master: |   at master: |
|   $d^t := \frac{1}{n}\sum_{i=1}^n d_i^t$ |   $d^t := \frac{1}{n}\sum_{i=1}^n d_i^t$ |   $d^t := \frac{1}{n}\sum_{i=1}^n d_i^t$ |
|   $h^{t+1} := h^t + \lambda d^t$ |   $h^{t+1} := h^t + d^t$ |   $h^{t+1} := h^t + \lambda d^t$ |
|   $g^{t+1} := h^t + \nu d^t$ |   $g^{t+1} := h^t + d^t$ |   $g^{t+1} := h^t + d^t$ |
|   $x^{t+1} := \text{prox}_{\gamma R}(x^t - \gamma g^{t+1})$ |   $x^{t+1} := \text{prox}_{\gamma R}(x^t - \gamma g^{t+1})$ |   $x^{t+1} := \text{prox}_{\gamma R}(x^t - \gamma g^{t+1})$ |
|   broadcast $x^{t+1}$ to all workers |   broadcast $x^{t+1}$ to all workers |   broadcast $x^{t+1}$ to all workers |
| **end for** | **end for** | **end for** |

Figure 1: In the three algorithms, $g^{t+1}$ is an estimate of $\nabla f(x^t)$, the $h_i^t$ are control variates converging to $\nabla f_i(x^\star)$, and their average $h^t = \frac{1}{n}\sum_{i=1}^n h_i^t$ is maintained and updated by the master. EF21 is a particular case of EF-BV, when $\nu = \lambda = 1$ and the compressors are in $\mathbb{B}(\alpha)$; then $g^{t+1}$ is simply equal to $h^{t+1}$ for every $t \geq 0$. DIANA is a particular case of EF-BV, when $\nu = 1$ and the compressors are in $\mathbb{U}(\omega)$; then $g^t$ is an unbiased estimate of $\nabla f(x^t)$.

    2. If the scaled compressors $\lambda\mathcal{C}_i^t$ are in $\mathbb{B}(\alpha)$, for some $\alpha \in (0,1]$ and $\lambda \in (0,1)$ (see Proposition 2), one can simply use these scaled compressors in EF21. This is equivalent to using EF-BV with the original compressors $\mathcal{C}_i^t$, the scaling with $\lambda$ taking place inside the algorithm. But we must have $\nu = \lambda$ for this equivalence to hold.

Therefore, we consider thereafter that EF21 corresponds to the particular case of EF-BV with $\nu = \lambda \in (0,1]$ and $\lambda\mathcal{C}_i^t \in \mathbb{B}(\alpha)$, for some $\alpha \in (0,1]$, and is not only the original algorithm shown in Fig. 1, which has no scaling parameter (but scaling might have been applied beforehand to make the compressors in $\mathbb{B}(\alpha)$).

### 3.2   DIANA **as a particular case of** EF-BV

EF-BV with $\nu = 1$ yields exactly DIANA, as shown in Fig. 1. DIANA was only studied with unbiased compressors $\mathcal{C}_i^t \in \mathbb{U}(\omega)$, for some $\omega \geq 0$. In that case, $\mathbb{E}\big[g^{t+1}\big] = \nabla f(x^t)$, so that $g^{t+1}$ is an unbiased stochastic gradient estimate; this is not the case in EF21 and EF-BV, in general. Also, $\lambda = \frac{1}{1+\omega}$ is the usual choice in DIANA, which is consistent with Proposition 2.

## 4   Linear convergence results

We will prove linear convergence of EF-BV under conditions weaker than strong convexity of $f + R$.

When $R = 0$, we will consider the Polyak–Łojasiewicz (PŁ) condition on $f$: $f$ is said to satisfy the PŁ condition with constant $\mu > 0$ if, for every $x \in \mathbb{R}^d$, $\|\nabla f(x)\|^2 \geq 2\mu\big(f(x) - f^\star\big)$, where $f^\star = f(x^\star)$, for any minimizer $x^\star$ of $f$. This holds if, for instance, $f$ is $\mu$-strongly convex; that is, $f - \frac{\mu}{2}\|\cdot\|^2$ is convex. In the general case, we will consider the Kurdyka–Łojasiewicz (KŁ) condition with exponent $1/2$ (Attouch & Bolte, 2009; Karimi et al., 2016) on $f + R$: $f + R$ is said to satisfy the KŁ condition with constant $\mu > 0$ if, for every $x \in \mathbb{R}^d$ and $u \in \partial R(x)$,

$$\|\nabla f(x) + u\|^2 \geq 2\mu\big(f(x) + R(x) - f^\star - R^\star\big), \tag{7}$$

where $f^\star = f(x^\star)$ and $R^\star = R(x^\star)$, for any minimizer $x^\star$ of $f + R$. This holds if, for instance, $R = 0$ and $f$ satisfies the PŁ condition with constant $\mu$, so that the KŁ condition generalizes the PŁ

condition to the general case $R \neq 0$. The KŁ condition also holds if $f + R$ is $\mu$-strongly convex (Karimi et al., 2016), for which it is sufficient that $f$ is $\mu$-strongly convex, or $R$ is $\mu$-strongly convex.

In the rest of this section, we assume that $\mathcal{C}_i^t \in \mathbb{C}(\eta, \omega)$, for some $\eta \in [0, 1)$ and $\omega \geq 0$, and we introduce $\omega_{\mathrm{av}} \leq \omega$ such that (6) is satisfied. According to the discussion in Sect. 2.5 (see also Remark 1 below), we define the optimal values for the scaling parameters $\lambda$ and $\nu$:

$$\lambda^\star := \min\left(\frac{1-\eta}{(1-\eta)^2 + \omega}, 1\right), \qquad \nu^\star := \min\left(\frac{1-\eta}{(1-\eta)^2 + \omega_{\mathrm{av}}}, 1\right).$$

Given $\lambda \in (0, 1]$ and $\nu \in (0, 1]$, we define for convenience $r := (1 - \lambda + \lambda\eta)^2 + \lambda^2\omega$, $r_{\mathrm{av}} := (1 - \nu + \nu\eta)^2 + \nu^2\omega_{\mathrm{av}}$, as well as $s^\star := \sqrt{\frac{1+r}{2r}} - 1$ and $\theta^\star := s^\star(1 + s^\star)\frac{r}{r_{\mathrm{av}}}$.

Note that if $r < 1$, according to Proposition 1 and (5), $\lambda\mathcal{C}_i^t \in \mathbb{B}(\alpha)$, with $\alpha = 1 - r$.

Our linear convergence results for EF-BV are the following:

**Theorem 1.** *Suppose that $R = 0$ and $f$ satisfies the PŁ condition with some constant $\mu > 0$. In* EF-BV, *suppose that $\nu \in (0, 1]$, $\lambda \in (0, 1]$ is such that $r < 1$, and*

$$0 < \gamma \leq \frac{1}{L + \tilde{L}\sqrt{\frac{r_{\mathrm{av}}}{r}}\frac{1}{s^\star}}. \tag{8}$$

*For every $t \geq 0$, define the Lyapunov function $\Psi^t := f(x^t) - f^\star + \frac{\gamma}{2\theta^\star}\frac{1}{n}\sum_{i=1}^n \left\|\nabla f_i(x^t) - h_i^t\right\|^2$,*

*where $f^\star := f(x^\star)$, for any minimizer $x^\star$ of $f$. Then, for every $t \geq 0$,*

$$\mathbb{E}\big[\Psi^t\big] \leq \left(\max\left(1 - \gamma\mu, \frac{r+1}{2}\right)\right)^t \Psi^0. \tag{9}$$

**Theorem 2.** *Suppose that $f + R$ satisfies the the KŁ condition with some constant $\mu > 0$. In* EF-BV, *suppose that $\nu \in (0, 1]$, $\lambda \in (0, 1]$ is such that $r < 1$, and*

$$0 < \gamma \leq \frac{1}{2L + \tilde{L}\sqrt{\frac{r_{\mathrm{av}}}{r}}\frac{1}{s^\star}}. \tag{10}$$

*$\forall t \geq 0$, define the Lyapunov function $\Psi^t := f(x^t) + R(x^t) - f^\star - R^\star + \frac{\gamma}{2\theta^\star}\frac{1}{n}\sum_{i=1}^n \left\|\nabla f_i(x^t) - h_i^t\right\|^2$,*

*where $f^\star := f(x^\star)$ and $R^\star := R(x^\star)$, for any minimizer $x^\star$ of $f + R$. Then, for every $t \geq 0$,*

$$\mathbb{E}\big[\Psi^t\big] \leq \left(\max\left(\frac{1}{1 + \frac{1}{2}\gamma\mu}, \frac{r+1}{2}\right)\right)^t \Psi^0. \tag{11}$$

**Remark 1** (choice of $\lambda, \nu, \gamma$ in EF-BV)**.** In Theorems 1 and 2, the rate is better if $r$ is small and $\gamma$ is large. So, we should take $\gamma$ equal to the upper bound in (8) and (10), since there is no reason to choose it smaller. Also, this upper bound is large if $r$ and $r_{\mathrm{av}}$ are small. As discussed in Sect. 2.5, $r$ and $r_{\mathrm{av}}$ are minimized with $\lambda = \lambda^\star$ and $\nu = \nu^\star$ (which implies that $r_{\mathrm{av}} \leq r < 1$), so this is the recommended choice. Also, with this choice of $\lambda, \nu, \gamma$, there is no parameter left to tune in the algorithm, which is a nice feature.

**Remark 2** (low noise regime)**.** *When the compression error tends to zero, i.e. $\eta$ and $\omega$ tend to zero, and we use accordingly $\lambda \to 1$, $\nu \to 1$, such that $r_{\mathrm{av}}/r$ remains bounded, then $\mathcal{C}_i^t \to \mathrm{Id}$, $r \to 0$, and $\frac{1}{s^\star} \to 0$. Hence,* EF-BV *reverts to proximal gradient descent $x^{t+1} = \mathrm{prox}_{\gamma R}\big(x^t - \nabla f(x^t)\big)$.*

**Remark 3** (high noise regime)**.** *When the compression error becomes large, i.e. $\eta \to 1$ or $\omega \to +\infty$, then $r \to 1$ and $\frac{1}{s^\star} \sim \frac{4}{1-r}$. Hence, the asymptotic complexity of* EF-BV *to achieve $\epsilon$-accuracy, when $\gamma = \Theta\big(\frac{1}{L + \tilde{L}\sqrt{\frac{r_{\mathrm{av}}}{r}}\frac{1}{s^\star}}\big)$, is*

$$\mathcal{O}\left(\left(\frac{L}{\mu} + \left(\frac{\tilde{L}}{\mu}\sqrt{\frac{r_{\mathrm{av}}}{r}} + 1\right)\frac{1}{1-r}\right)\log\frac{1}{\epsilon}\right). \tag{12}$$

## 4.1 Implications for EF21

Let us assume that $\nu = \lambda$, so that EF-BV reverts to EF21, as explained in Sect. 3.1. Then, if we don't assume the prior knowledge of $\omega_{\mathrm{av}}$, or equivalently if $\omega_{\mathrm{av}} = \omega$, Theorem 1 with $r = r_{\mathrm{av}}$ recovers the linear convergence result of EF21 due to Richtárik et al. (2021), up to slightly different constants.

However, in these same conditions, Theorem 2 is new: linear convergence of EF21 with $R \neq 0$ was only shown in Theorem 13 of Fatkhullin et al. (2021), under the assumption that there exists $\mu > 0$, such that for every $x \in \mathbb{R}^d$, $\frac{1}{\gamma^2} \left\| x - \mathrm{prox}_{\gamma R}\big(x - \gamma \nabla f(x)\big) \right\|^2 \geq 2\mu\big(f(x) + R(x) - f^\star - R^\star\big)$. This condition generalizes the PŁ condition, since it reverts to it when $R = 0$, but it is different from the KŁ condition, and it is not clear when it is satisfied, in particular whether it is implied by strong convexity of $f + R$.

The asymptotic complexity to achieve $\epsilon$-accuracy of EF21 with $\gamma = \Theta\big(\frac{1}{L + \tilde{L}/s^\star}\big)$ is $\mathcal{O}\big(\frac{\tilde{L}}{\mu} \frac{1}{1-r} \log \frac{1}{\epsilon}\big)$ (where we recall that $1 - r = \alpha$, with the scaled compressors in $\mathbb{B}(\alpha)$). Thus, for a given problem and compressors, the improvement of EF-BV over EF21 is the factor $\sqrt{\frac{r_{\mathrm{av}}}{r}}$ in (12), which can be small if $n$ is large.

Theorems 1 and 2 provide a new insight about EF21: if we exploit the knowledge that $\mathcal{C}_i^t \in \mathbb{C}(\eta, \omega)$ and the corresponding constant $\omega_{\mathrm{av}}$, and if $\omega_{\mathrm{av}} < \omega$, then $r_{\mathrm{av}} < r$, so that, based on (8) and (10), $\gamma$ can be chosen larger than with the default assumption that $r_{\mathrm{av}} = r$. As a consequence, convergence will be faster. This illustrates the interest of our new finer parameterization of compressors with $\eta, \omega, \omega_{\mathrm{av}}$. However, it is only half the battle to make use of the factor $\frac{r_{\mathrm{av}}}{r}$ in EF21: the property $\omega_{\mathrm{av}} < \omega$ is only really exploited if $\nu = \nu^\star$ in EF-BV (since $r_{\mathrm{av}}$ is minimized this way). In other words, there is no reason to set $\nu = \lambda$ in EF-BV, when a larger value of $\nu$ is allowed in Theorems 1 and 2 and yields faster convergence.

## 4.2 Implications for DIANA

Let us assume that $\nu = 1$, so that EF-BV reverts to DIANA, as explained in Sect. 3.2. This choice is allowed in Theorems 1 and 2, so that they provide new convergence results for DIANA. Assuming that the compressors are unbiased, i.e. $\mathcal{C}_i^t \in \mathbb{U}(\omega)$ for some $\omega \geq 0$, we have the following result on DIANA (Condat & Richtárik, 2022, Theorem 5 with $b = \sqrt{2}$):

**Proposition 3.** *Suppose that $f$ is $\mu$-strongly convex, for some $\mu > 0$, and that in DIANA, $\lambda = \frac{1}{1+\omega}$, $0 < \gamma \leq \frac{1}{L_{\max} + L_{\max}(1+\sqrt{2})^2 \omega_{\mathrm{av}}}$. For every $t \geq 0$, define the Lyapunov function $\Phi^t := \|x^t - x^\star\|^2 + (2 + \sqrt{2})\gamma^2 \omega_{\mathrm{av}}(1 + \omega)\frac{1}{n}\sum_{i=1}^n \|\nabla f_i(x^\star) - h_i^t\|^2$, where $x^\star$ is the minimizer of $f + R$, which exists and is unique. Then, for every $t \geq 0$, $\mathbb{E}[\Phi^t] \leq \left(\max\left(1 - \gamma\mu, \frac{\frac{1}{2}+\omega}{1+\omega}\right)\right)^t \Phi^0$.*

Thus, noting that $r = \frac{\omega}{1+\omega}$, so that $\frac{r+1}{2} = \frac{\frac{1}{2}+\omega}{1+\omega}$, the rate is exactly the same as in Theorem 1, but with a different Lyapunov function. Theorems 1 and 2 have the advantage over Proposition 3, that linear convergence is guaranteed under the PŁ or KŁ assumptions, which are weaker than strong convexity of $f$. Also, the constants $L$ and $\tilde{L}$ appear instead of $L_{\max}$. This shows a better dependence with respect to the problem. However, noting that $r = \frac{\omega}{1+\omega}$, $r_{\mathrm{av}} = \omega_{\mathrm{av}}$, $\frac{1}{s^\star} \sim 4\omega$, the factor $\sqrt{\frac{r_{\mathrm{av}}}{r}} \frac{1}{s^\star}$ scales like $\sqrt{\omega_{\mathrm{av}}}\omega$, which is worse that $\omega_{\mathrm{av}}$. This means that $\gamma$ can certainly be chosen larger in Proposition 3 than in Theorems 1 and 2, leading to faster convergence.

However, Theorems 1 and 2 bring a major highlight: for the first time, they establish convergence of DIANA, which is EF-BV with $\nu = 1$, with biased compressors. We state the results in Appendix B, by lack of space. In any case, with biased compressors, it is better to use EF-BV than DIANA: there is no interest in choosing $\nu = 1$ instead of $\nu = \nu^\star$, which minimizes $r_{\mathrm{av}}$ and allows for a larger $\gamma$, for faster convergence.

Finally, we can remark that for unbiased compressors with $\omega_{\mathrm{av}} \ll 1$, for instance if $\omega_{\mathrm{av}} \approx \frac{\omega}{n}$ with $n$ larger than $\omega$, then $\nu^\star = \frac{1}{1+\omega_{\mathrm{av}}} \approx 1$. Thus, in this particular case, EF-BV with $\nu = \nu^\star$ and DIANA are essentially the same algorithm. This is another sign that EF-BV with $\lambda = \lambda^\star$ and $\nu = \nu^\star$ is a generic and robust choice, since it recovers EF21 and DIANA in settings where these algorithms shine.

# 5   Sublinear convergence in the nonconvex case

In this section, we consider the general nonconvex setting. In (1), every function $f_i$ is supposed $L_i$-smooth, for some $L_i > 0$. For simplicity, we suppose that $R = 0$. As previously, we set $\tilde{L} := \sqrt{\frac{1}{n}\sum_{i=1}^n L_i^2}$. The average function $f := \frac{1}{n}\sum_{i=1}^n f_i$ is $L$-smooth, for some $L \leq \tilde{L}$. We also suppose that $f$ is bounded from below; that is, $f^{\text{inf}} := \inf_{x \in \mathbb{R}^d} f(x) > -\infty$.

Given $\lambda \in (0,1]$ and $\nu \in (0,1]$, we define for convenience $r := (1 - \lambda + \lambda\eta)^2 + \lambda^2\omega$, $r_{\text{av}} := (1 - \nu + \nu\eta)^2 + \nu^2\omega_{\text{av}}$, as well as $s := \frac{1}{\sqrt{r}} - 1$ and $\theta := s(1+s)\frac{r}{r_{\text{av}}}$. Our convergence result is the following:

**Theorem 3.** *In* EF-BV, *suppose that* $\nu \in (0,1]$, $\lambda \in (0,1]$ *is such that* $r < 1$, *and*

$$0 < \gamma \leq \frac{1}{L + \tilde{L}\sqrt{\frac{r_{\text{av}}}{r}}\frac{1}{s}}. \tag{13}$$

*For every* $t \geq 1$, *let* $\hat{x}^t$ *be chosen from the iterates* $x^0, x^1, \cdots, x^{t-1}$ *uniformly at random. Then*

$$\mathbb{E}\left[\left\|\nabla f(\hat{x}^t)\right\|^2\right] \leq \frac{2\big(f(x^0) - f^{\text{inf}}\big)}{\gamma t} + \frac{G^0}{\theta t}, \tag{14}$$

*where* $G^0 := \frac{1}{n}\sum_{i=1}^n \left\|\nabla f_i(x^0) - h_i^0\right\|^2$.

# 6   Experiments

We conducted comprehensive experiments to illustrate the efficiency of EF-BV compared to EF21 (we use biased compressors, so we don't include DIANA in the comparison). The settings and results are detailed in Appendix C and some results are shown in Fig. 2; we can see the speedup obtained with EF-BV, which exploits the randomness of the compressors.

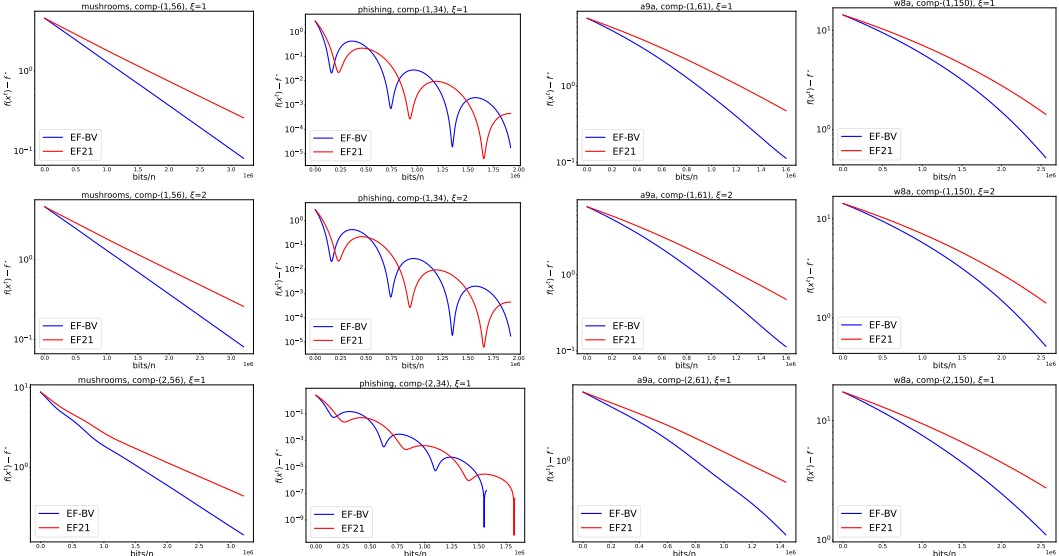

Figure 2: Experimental results. We plot $f(x^t) - f^\star$ with respect to the number of bits sent by each node during the learning process, which is proportional to $tk$. Top row: comp-$(1, d/2)$, overlapping $\xi = 1$. Middle row: comp-$(1, d/2)$, overlapping $\xi = 2$. Bottom row: comp-$(2, d/2)$, overlapping $\xi = 1$.

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
