# Appendix

## Contents

# A  New compressors

We propose new compressors in our class $\mathbb{C}(\eta, \omega)$.

## A.1  mix-(k,k'): Mixture of top-k and rand-k

Let $k \in \mathcal{I}_d$ and $k' \in \mathcal{I}_d$, with $k + k' \leq d$. We propose the compressor mix-$(k, k')$. It maps $x \in \mathbb{R}^d$ to $x' \in \mathbb{R}^d$, defined as follows. Let $i_1, \ldots, i_k$ be distinct indexes in $\mathcal{I}_d$ such that $|x_{i_1}|, \ldots, |x_{i_k}|$ are the $k$ largest elements of $|x|$ (if this selection is not unique, we can choose any one). These coordinates are kept: $x'_{i_j} = x_{i_j}$, $j = 1, \ldots, k$. In addition, $k'$ other coordinates chosen at random in the remaining ones are kept: $x'_{i_j} = x_{i_j}$, $j = k+1, \ldots, k+k'$, where $\{i_j : j = k+1, \ldots, k+k'\}$ is a subset of size $k'$ of $\mathcal{I}_d \backslash \{i_1, \ldots, i_k\}$ chosen uniformly at random. The other coordinates of $x'$ are set to zero.

**Proposition 4.** mix-$(k, k') \in \mathbb{C}(\eta, \omega)$ with $\eta = \frac{d-k-k'}{\sqrt{(d-k)d}}$ and $\omega = \frac{k'(d-k-k')}{(d-k)d}$.

As a consequence, mix-$(k, k') \in \mathbb{B}(\alpha)$ with $\alpha = 1 - \eta^2 - \omega = 1 - \frac{(d-k-k')^2}{(d-k)d} - \frac{k'(d-k-k')}{(d-k)d} = \frac{k+k'}{d}$. This is the same $\alpha$ as for top-$(k + k')$ and scaled rand-$(k + k')$.

The proof is given in Appendix D.

## A.2  comp-(k,k'): Composition of top-k and rand-k

Let $k \in \mathcal{I}_d$ and $k' \in \mathcal{I}_d$, with $k \leq k'$. We consider the compressor comp-$(k, k')$, proposed in Barnes et al. (2020), which is the composition of top-$k'$ and rand-$k$: top-$k'$ is applied first, then rand-$k$ is applied to the $k'$ selected (largest) elements. That is, comp-$(k, k')$ maps $x \in \mathbb{R}^d$ to $x' \in \mathbb{R}^d$, defined as follows. Let $i_1, \ldots, i_{k'}$ be distinct indexes in $\mathcal{I}_d$ such that $|x_{i_1}|, \ldots, |x_{i_{k'}}|$ are the $k'$ largest elements of $|x|$ (if this selection is not unique, we can choose any one). Then $x'_{i_j} = \frac{k'}{k} x_{i_j}$, $j = 1, \ldots, k$, where $\{i_j : j = 1, \ldots, k\}$ is a subset of size $k$ of $\{i_1, \ldots, i_{k'}\}$ chosen uniformly at random. The other coordinates of $x'$ are set to zero.

comp-$(k, k')$ sends $k$ coordinates of its input vector, like top-$k$ and rand-$k$, whatever $k'$. We can note that comp-$(k, d) =$ rand-$k$ and comp-$(k, k) =$ top-$k$. We have:

**Proposition 5.** comp-$(k, k') \in \mathbb{C}(\eta, \omega)$ with $\eta = \sqrt{\frac{d-k'}{d}}$ and $\omega = \frac{k'-k}{k}$.

The proof is given in Appendix E.

# B  New results on DIANA

We suppose that the compressors $\mathcal{C}_i^t$ are in $\mathbb{C}(\eta, \omega)$, for some $\eta \in [0, 1)$ and $\omega \geq 0$. Viewing DIANA as EF-BV with $\nu = 1$, we define $r$, $s^\star$, $\theta^\star$ as before, as well as $r_{\mathrm{av}} := \eta^2 + \omega_{\mathrm{av}}$. We obtain, as corollaries of Theorems 1 and 2:

**Theorem 4.** *Suppose that $R = 0$ and $f$ satisfies the PŁ condition with some constant $\mu > 0$. In DIANA, suppose that $\lambda \in (0, 1]$ is such that $r < 1$, and*

$$0 < \gamma \leq \frac{1}{L + \tilde{L}\sqrt{\frac{r_{\mathrm{av}}}{r}\frac{1}{s^\star}}}.$$

*For every $t \geq 0$, define the Lyapunov function $\Psi^t := f(x^t) - f^\star + \frac{\gamma}{2\theta^\star}\frac{1}{n}\sum_{i=1}^n \|\nabla f_i(x^t) - h_i^t\|^2$, where $f^\star := f(x^\star)$, for any minimizer $x^\star$ of $f$. Then, for every $t \geq 0$,*

$$\mathbb{E}\big[\Psi^t\big] \leq \left(\max\left(1 - \gamma\mu, \frac{r+1}{2}\right)\right)^t \Psi^0.$$

**Theorem 5.** *Suppose that $f + R$ satisfies the the KŁ condition with some constant $\mu > 0$. In DIANA, suppose that $\lambda \in (0, 1]$ is such that $r < 1$, and*

$$0 < \gamma \leq \frac{1}{2L + \tilde{L}\sqrt{\frac{r_{\mathrm{av}}}{r}\frac{1}{s^\star}}}.$$

$\forall t \geq 0$, *define the Lyapunov function* $\Psi^t := f(x^t) + R(x^t) - f^\star - R^\star + \frac{\gamma}{2\theta^\star} \frac{1}{n} \sum_{i=1}^n \|\nabla f_i(x^t) - h_i^t\|^2$, *where* $f^\star := f(x^\star)$ *and* $R^\star := R(x^\star)$, *for any minimizer* $x^\star$ *of* $f + R$. *Then, for every* $t \geq 0$,

$$\mathbb{E}[\Psi^t] \leq \left( \max \left( \frac{1}{1 + \frac{1}{2}\gamma\mu}, \frac{r+1}{2} \right) \right)^t \Psi^0.$$

Interestingly, DIANA, used beyond its initial setting with compressors in $\mathbb{B}(\alpha)$ with $\lambda = 1$, just reverts to (the original) EF21, as shown in Fig. 1. This shows how our unified framework reveals connections between these two algorithms and unleashes their potential.

## C  Experiments

### C.1  Datasets and experimental setup

We consider the heterogeneous data distributed regime, which means that all parallel nodes store different data points, but use the same type of learning function. We adopt the datasets from LibSVM (Chang & Lin, 2011) and we split them, after random shuffling, into $n \leq N$ blocks, where $N$ is the total number of data points (the left-out data points from the integer division of $N$ by $n$ are stored at the last node). The corresponding values are shown in Tab. 2. To make our setting more realistic, we consider that different nodes partially share some data: we set the overlapping factor to be $\xi \in \{1, 2\}$, where $\xi = 1$ means no overlap and $\xi = 2$ means that the data is partially shared among the nodes, with a redundancy factor of 2; this is achieved by sequentially assigning 2 blocks of data to every node. The experiments were conducted using 24 NVIDIA-A100-80G GPUs, each with 80GB memory.

We consider logistic regression, which consists in minimizing the $\mu$-strongly convex function

$$f = \frac{1}{n} \sum_{i=1}^n f_i,$$

with, for every $i \in \mathcal{I}_n$,

$$f_i(x) = \frac{1}{N_i} \sum_{j=1}^{N_i} \log \left( 1 + \exp\left(-b_{i,j} x^\top a_{i,j}\right) \right) + \frac{\mu}{2} \|x\|^2,$$

where $\mu$, set to $0.1$, is the strong convexity constant; $N_i$ is the number of data points at node $i$; the $a_{i,j}$ are the training vectors and the $b_{i,j} \in \{-1, 1\}$ the corresponding labels. Note that there is no regularizer in this problem; that is, $R = 0$.

We set $L = \tilde{L} = \sqrt{\sum_{i=1}^n L_i^2}$, with $L_i = \mu + \frac{1}{4N_i} \sum_{j=1}^{N_i} \|a_{i,j}\|^2$. We use independent compressors of type comp-$(k, k')$ at every node, for some small $k$ and large $k' < d$. These compressors are biased ($\eta > 0$) and have a variance $\omega > 1$, so they are not contractive: they don't belong to $\mathbb{B}(\alpha)$ for any $\alpha$. We have $\omega_{\text{av}} = \frac{\omega}{n}$. Thus, we place ourselves in the conditions of Theorem 1, and we compare EF-BV with

$$\lambda = \lambda^\star, \quad \nu = \nu^\star, \quad \gamma = \frac{1}{L + \tilde{L}\sqrt{\frac{r_{\text{av}}}{r} \frac{1}{s^\star}}}$$

to EF21, which corresponds to the particular case of EF-BV with

$$\nu = \lambda = \lambda^\star, \quad \gamma = \frac{1}{L + \tilde{L}\frac{1}{s^\star}}.$$

Table 2: Values of $d$ and $N$ for the considered datasets.

| Dataset | $N$ (total # of datapoints) | $d$ (# of features) |
|---|---|---|
| mushrooms | 8,124 | 112 |
| phishing | 11,055 | 68 |
| a9a | 32,561 | 123 |
| w8a | 49,749 | 300 |

Table 3: Parameter values of EF-BV and EF21 in the different settings. $k'$ in comp-$(k, k')$ is set to $d/2$ and $n = 1000$. In pairs of values like (1,2), the first value is $k$ and the second value is $\xi$.

| Method | Params | mushrooms | | | phishing | | | a9a | | | w8a | | |
|---|---|---|---|---|---|---|---|---|---|---|---|---|---|
| | | (1,1) | (1,2) | (2,1) | (1,1) | (1,2) | (2,1) | (1,1) | (1,2) | (2,1) | (1,1) | (1,2) | (2,1) |
| | $\eta$ | 0.707 | 0.707 | 0.707 | 0.707 | 0.707 | 0.707 | 0.710 | 0.710 | 0.710 | 0.707 | 0.707 | 0.707 |
| | $\omega$ | 55 | 55 | 27 | 33 | 33 | 16 | 60 | 60 | 29.5 | 149 | 149 | 74 |
| | $\omega_{\mathrm{av}}$ | 0.055 | 0.055 | 0.027 | 0.033 | 0.033 | 0.016 | 0.06 | 0.06 | 0.295 | 0.149 | 0.149 | 0.074 |
| EF-BV | $\lambda$ | 5.32e-3 | 5.32e-3 | 1.08e-2 | 8.85e-3 | 8.85e-3 | 1.82e-2 | 4.83e-3 | 4.83e-3 | 9.8e-3 | 1.96e-3 | 1.96e-3 | 3.95e-3 |
| EF21 | | 5.32e-3 | 5.32e-4 | 1.08e-2 | 8.85e-3 | 8.85e-3 | 1.82e-2 | 4.83e-3 | 4.83e-3 | 9.8e-3 | 1.96e-3 | 1.96e-3 | 3.95e-3 |
| EF-BV | $\nu$ | 1 | 1 | 1 | 1 | 1 | 1 | 1 | 1 | 1 | 1 | 1 | 1 |
| EF21 | | 5.32e-3 | 5.32e-4 | 1.08e-2 | 8.85e-3 | 8.85e-3 | 1.82e-2 | 4.83e-3 | 4.83e-3 | 9.8e-3 | 1.96e-3 | 1.96e-3 | 3.95e-3 |
| EF-BV | $r$ | 0.998 | 0.998 | 0.997 | 0.997 | 0.997 | 0.994 | 0.999 | 0.999 | 0.997 | 0.999 | 0.999 | 0.999 |
| EF21 | | 0.998 | 0.998 | 0.997 | 0.997 | 0.997 | 0.994 | 0.999 | 0.999 | 0.997 | 0.999 | 0.999 | 0.999 |
| EF-BV | $r_{\mathrm{av}}$ | 0.555 | 0.555 | 0.527 | 0.533 | 0.533 | 0.516 | 0.564 | 0.564 | 0.534 | 0.649 | 0.649 | 0.574 |
| EF21 | | 0.998 | 0.998 | 0.997 | 0.997 | 0.997 | 0.994 | 0.999 | 0.999 | 0.997 | 0.999 | 0.999 | 0.999 |
| EF-BV | $\sqrt{\frac{r_{\mathrm{av}}}{r}}$ | 0.746 | 0.746 | 0.727 | 0.731 | 0.731 | 0.720 | 0.752 | 0.752 | 0.731 | 0.806 | 0.806 | 0.758 |
| EF21 | | 1 | 1 | 1 | 1 | 1 | 1 | 1 | 1 | 1 | 1 | 1 | 1 |
| EF-BV | $s^\star$ | 3.90e-4 | 3.90e-4 | 7.94e-4 | 6.50e-4 | 6.50e-4 | 1.34e-3 | 3.5e-4 | 3.5e-4 | 7.13e-4 | 1.44e-4 | 1.44e-4 | 2.90e-4 |
| EF21 | | 3.90e-4 | 3.90e-4 | 7.94e-4 | 6.50e-4 | 6.50e-4 | 1.34e-3 | 3.5e-4 | 3.5e-4 | 7.13e-4 | 1.44e-4 | 1.44e-4 | 2.90e-4 |
| EF-BV | $\gamma$ | 1.38e-4 | 1.43e-4 | 2.87e-4 | 2.33e-3 | 2.36e-3 | 4.80e-3 | 2.53e-4 | 2.58e-4 | 5.28e-4 | 1.01e-4 | 1.15e-4 | 2.15e-4 |
| EF21 | | 1.03e-4 | 1.06e-4 | 2.10e-4 | 1.71e-3 | 1.73e-3 | 3.49e-3 | 1.91e-4 | 1.84e-4 | 3.87e-4 | 8.12e-5 | 9.31e-5 | 1.63e-4 |

## C.2 Experimental results and analysis

We show in Fig. 2 the results with $k = 1$ or $k = 2$ in the compressors comp-$(k, k')$, and overlapping factor $\xi = 1$ or $\xi = 2$. We chose $k' = \frac{d}{2}$ and $n = 1000$. The corresponding values of $\eta, \omega, \omega_{\mathrm{av}}$, and the parameter values used in the algorithms are shown in Tab. 3. We can see that there is essentially no difference between the two choices $\xi = 1$ and $\xi = 2$, and the qualitative behavior for $k = 1$ and $k = 2$ is similar. Thus, we observe that EF-BV converges always faster than EF21; this is consistent with our analysis.

We tried other values of $n$, including the largest value $n = N$, for which there is only one data point at every node. The behavior of EF21 and EF-BV was the same as for $n = 1000$, so we don't show the results.

We tried other values of $k'$. The behavior of EF21 and EF-BV was the same as for $k' = \frac{d}{2}$ overall, so we don't show the results. We noticed that the difference between the two algorithms was smaller when $k'$ was smaller; this is expected, since for $k' = k$, the compressors revert to top-$k$, for which EF21 and EF-BV are the same algorithm.

To sum up, the experiments confirm our analysis: when $\omega$ and $n$ are large, so that the key factor $\sqrt{\frac{r_{\mathrm{av}}}{r}}$ is small, randomness is exploited in EF-BV, with larger values of $\nu$ and $\gamma$ allowed than in EF21, and this yields faster convergence.

In future work, we will design and compare other compressors in our new class $\mathbb{C}(\eta, \omega)$, performing well in both homogeneous and heterogeneous regimes.

## C.3 Additional experiments in the nonconvex setting

We consider the logistic regression problem with a nonconvex regularizer:

$$f(x) = \frac{1}{n} \sum_{i=1}^{n} \log\left(1 + \exp\left(-y_i a_i^\top x\right)\right) + \lambda \sum_{j=1}^{d} \frac{x_j^2}{1 + x_j^2}, \tag{15}$$

where $a_i \in \mathbb{R}^d$, $y_i \in \{-1, 1\}$ are the training data, and $\lambda > 0$ is the regularizer parameter. We used $\lambda = 0.1$ in all experiments. We present the results in Fig. 3.

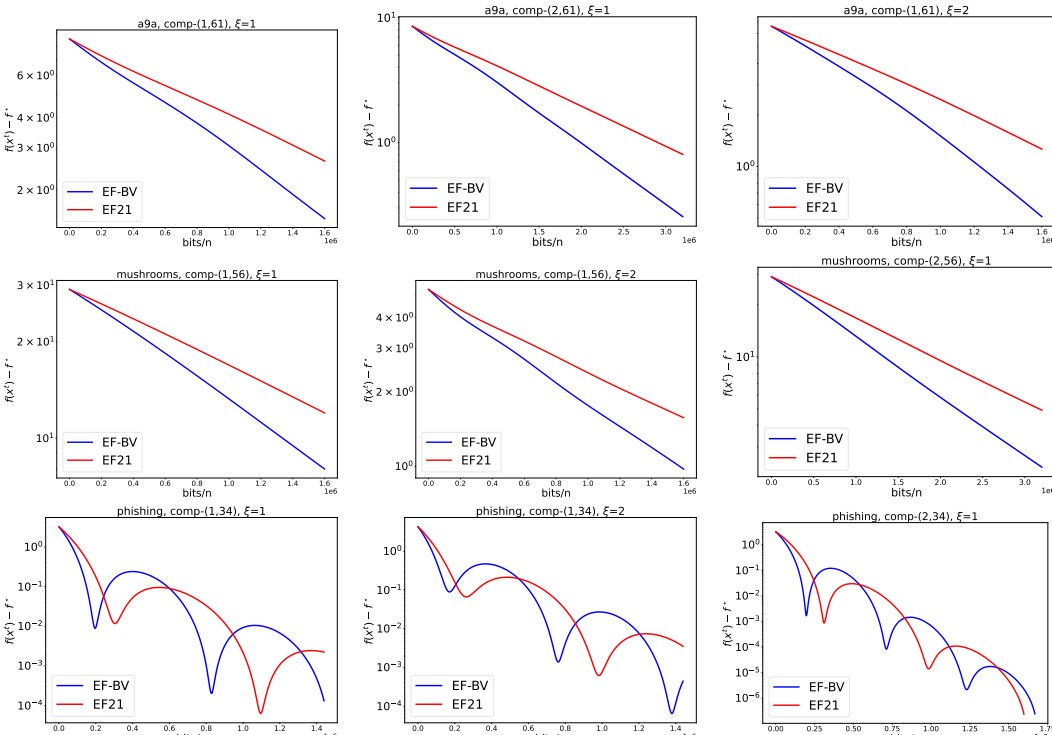

Figure 3: Comparison between EF21 and EF-BV in the nonconvex setting. We see that EF-BV outperforms EF21 for all datasets.

# D Proof of Proposition 4

We first calculate $\omega$. Let $x \in \mathbb{R}^d$.

$$\left\|\mathcal{C}(x) - \mathbb{E}[\mathcal{C}(x)]\right\|^2 = \sum_{i \in \mathcal{I}_d \setminus \{i_1, \ldots, i_{k+k'}\}} \left(\frac{k'}{d-k}\right)^2 |x_i|^2 + \sum_{j=k+1}^{k+k'} \left(\frac{d-k-k'}{d-k}\right)^2 |x_{i_j}|^2.$$

Therefore, by taking the expectation over the random indexes $i_{k+1}, \ldots, i_{2k}$,

$$\mathbb{E}\left[\left\|\mathcal{C}(x) - \mathbb{E}[\mathcal{C}(x)]\right\|^2\right] = \sum_{i \in \mathcal{I}_d \setminus \{i_1, \ldots, i_k\}} \left(\frac{d-k-k'}{d-k}\left(\frac{k'}{d-k}\right)^2 + \frac{k'}{d-k}\left(\frac{d-k-k'}{d-k}\right)^2\right) |x_i|^2$$

$$= \frac{k'(d-k-k')}{(d-k)^2} \sum_{i \in \mathcal{I}_d \setminus \{i_1, \ldots, i_k\}} |x_i|^2.$$

Moreover, since the $|x_{i_j}|$ are the largest elements of $|x|$, for every $j = 1, \ldots, k$,

$$|x_{i_j}|^2 \geq \frac{1}{d-k} \sum_{i \in \mathcal{I}_d \setminus \{i_1, \ldots, i_k\}} |x_i|^2,$$

so that

$$\|x\|^2 = \sum_{i \in \mathcal{I}_d} |x_i|^2 \geq \left(1 + \frac{k}{d-k}\right) \sum_{i \in \mathcal{I}_d \setminus \{i_1, \ldots, i_k\}} |x_i|^2.$$

Hence,

$$\mathbb{E}\left[\left\|\mathcal{C}(x) - \mathbb{E}[\mathcal{C}(x)]\right\|^2\right] \leq \frac{k'(d-k-k')}{(d-k)^2}\frac{d-k}{d}\|x\|^2 = \frac{k'(d-k-k')}{(d-k)d}\|x\|^2.$$

Then, let us calculate $\eta$.

$$\left\| \mathbb{E}[\mathcal{C}(x)] - x \right\|^2 = \sum_{i \in \mathcal{I}_d \setminus \{i_1, \dots, i_k\}} \left( \frac{d - k - k'}{d - k} \right)^2 |x_i|^2$$

$$\leq \frac{(d - k - k')^2}{(d - k)d} \|x\|^2.$$

Thus, $\eta = \frac{d - k - k'}{\sqrt{(d-k)d}}$.

## E Proof of Proposition 5

We first calculate $\omega$. Let $x \in \mathbb{R}^d$.

$$\left\| \mathcal{C}(x) - \mathbb{E}[\mathcal{C}(x)] \right\|^2 = \sum_{j \in \{j_1, \dots, j_k\}} \left( \frac{k' - k}{k} \right)^2 |x_{i_j}|^2 + \sum_{i \in \{i_1, \dots, i_{k'}\} \setminus \{i_{j_1}, \dots, i_{j_k}\}} |x_i|^2$$

Therefore, by taking the expectation over the random indexes $i_{j_1}, \dots, i_{j_k}$,

$$\mathbb{E}\left[ \left\| \mathcal{C}(x) - \mathbb{E}[\mathcal{C}(x)] \right\|^2 \right] = \sum_{j=1}^{k'} \left( \frac{k}{k'} \left( \frac{k' - k}{k} \right)^2 + \frac{k' - k}{k'} \right) |x_{i_j}|^2$$

$$= \frac{k' - k}{k} \sum_{j=1}^{k'} |x_{i_j}|^2$$

$$\leq \frac{k' - k}{k} \|x\|^2$$

Then, let us calculate $\eta$:

$$\left\| \mathbb{E}[\mathcal{C}(x)] - x \right\|^2 = \sum_{i \in \mathcal{I}_d \setminus \{i_1, \dots, i_{k'}\}} |x_i|^2 \leq \frac{d - k'}{d} \|x\|^2.$$

## F Proof of Theorem 1

We have the descent property (Richtárik et al., 2021, Lemma 4), for every $t \geq 0$,

$$f(x^{t+1}) - f^\star \leq f(x^t) - f^\star - \frac{\gamma}{2} \left\| \nabla f(x^t) \right\|^2 + \frac{\gamma}{2} \left\| g^{t+1} - \nabla f(x^t) \right\|^2$$

$$+ \left( \frac{L}{2} - \frac{1}{2\gamma} \right) \left\| x^{t+1} - x^t \right\|^2 \tag{16}$$

$$\leq (1 - \gamma\mu)\left( f(x^t) - f^\star \right) + \frac{\gamma}{2} \left\| g^{t+1} - \nabla f(x^t) \right\|^2 + \left( \frac{L}{2} - \frac{1}{2\gamma} \right) \left\| x^{t+1} - x^t \right\|^2.$$

Then, for every $t \geq 0$, conditionally on $x^t$, $h^t$ and $(h_i^t)_{i=1}^n$,

$$\mathbb{E}\left[\left\|g^{t+1} - \nabla f(x^t)\right\|^2\right] = \mathbb{E}\left[\left\|\frac{1}{n}\sum_{i=1}^n \left(h_i^t - \nabla f_i(x^t) + \nu \mathcal{C}_i^t\left(\nabla f_i(x^t) - h_i^t\right)\right)\right\|^2\right]$$

$$= \left\|\frac{1}{n}\sum_{i=1}^n \left(h_i^t - \nabla f_i(x^t) + \nu\mathbb{E}\left[\mathcal{C}_i^t\left(\nabla f_i(x^t) - h_i^t\right)\right]\right)\right\|^2$$

$$+ \nu^2\mathbb{E}\left[\left\|\frac{1}{n}\sum_{i=1}^n \left(\mathcal{C}_i^t\left(\nabla f_i(x^t) - h_i^t\right) - \mathbb{E}\left[\mathcal{C}_i^t\left(\nabla f_i(x^t) - h_i^t\right)\right]\right)\right\|^2\right]$$

$$\leq \left\|\frac{1}{n}\sum_{i=1}^n \left(h_i^t - \nabla f_i(x^t) + \nu\mathbb{E}\left[\mathcal{C}_i^t\left(\nabla f_i(x^t) - h_i^t\right)\right]\right)\right\|^2$$

$$+ \nu^2\frac{\omega_{\mathrm{av}}}{n}\sum_{i=1}^n \left\|\nabla f_i(x^t) - h_i^t\right\|^2,$$

where the last inequality follows from (6). In addition,

$$\left\|\frac{1}{n}\sum_{i=1}^n \left(h_i^t - \nabla f_i(x^t) + \nu\mathbb{E}\left[\mathcal{C}_i^t\left(\nabla f_i(x^t) - h_i^t\right)\right]\right)\right\|$$

$$\leq \left\|\frac{1}{n}\sum_{i=1}^n \left(\nu\left(h_i^t - \nabla f_i(x^t)\right) + \nu\mathbb{E}\left[\mathcal{C}_i^t\left(\nabla f_i(x^t) - h_i^t\right)\right]\right)\right\|$$

$$+ (1-\nu)\left\|\frac{1}{n}\sum_{i=1}^n \left(h_i^t - \nabla f_i(x^t)\right)\right\|$$

$$\leq \frac{\nu}{n}\sum_{i=1}^n \left\|h_i^t - \nabla f_i(x^t) + \mathbb{E}\left[\mathcal{C}_i^t\left(\nabla f_i(x^t) - h_i^t\right)\right]\right\|$$

$$+ \frac{1-\nu}{n}\sum_{i=1}^n \left\|h_i^t - \nabla f_i(x^t)\right\|$$

$$\leq \frac{\nu\eta}{n}\sum_{i=1}^n \left\|\nabla f_i(x^t) - h_i^t\right\| + \frac{1-\nu}{n}\sum_{i=1}^n \left\|\nabla f_i(x^t) - h_i^t\right\|$$

$$= \frac{1 - \nu + \nu\eta}{n}\sum_{i=1}^n \left\|\nabla f_i(x^t) - h_i^t\right\|.$$

Therefore,

$$\left\|\frac{1}{n}\sum_{i=1}^n \left(h_i^t - \nabla f_i(x^t) + \nu\mathbb{E}\left[\mathcal{C}_i^t\left(\nabla f_i(x^t) - h_i^t\right)\right]\right)\right\|^2 \leq \frac{(1 - \nu + \nu\eta)^2}{n}\sum_{i=1}^n \left\|\nabla f_i(x^t) - h_i^t\right\|^2,$$

and, conditionally on $x^t$, $h^t$ and $(h_i^t)_{i=1}^n$,

$$\mathbb{E}\left[\left\|g^{t+1} - \nabla f(x^t)\right\|^2\right] \leq \left((1 - \nu + \nu\eta)^2 + \nu^2\omega_{\mathrm{av}}\right)\frac{1}{n}\sum_{i=1}^n \left\|\nabla f_i(x^t) - h_i^t\right\|^2.$$

Thus, for every $t \geq 0$, conditionally on $x^t$, $h^t$ and $(h_i^t)_{i=1}^n$,

$$\mathbb{E}\left[f(x^{t+1}) - f^\star\right] \leq (1 - \gamma\mu)\left(f(x^t) - f^\star\right) + \frac{\gamma}{2}\left((1 - \nu + \nu\eta)^2 + \nu^2\omega_{\mathrm{av}}\right)\frac{1}{n}\sum_{i=1}^n \left\|\nabla f_i(x^t) - h_i^t\right\|^2$$

$$+ \left(\frac{L}{2} - \frac{1}{2\gamma}\right)\mathbb{E}\left[\left\|x^{t+1} - x^t\right\|^2\right].$$

Now, let us study the control variates $h_i^t$. Let $s > 0$. Using the Peter–Paul inequality $\|a + b\|^2 \leq (1+s)\|a\|^2 + (1+s^{-1})\|b\|^2$, for any vectors $a$ and $b$, we have, for every $t \geq 0$ and $i \in \mathcal{I}_n$,

$$
\begin{aligned}
\left\|\nabla f_i(x^{t+1}) - h_i^{t+1}\right\|^2 &= \left\|h_i^t - \nabla f_i(x^{t+1}) + \lambda \mathcal{C}_i^t\big(\nabla f_i(x^t) - h_i^t\big)\right\|^2 \\
&\leq (1+s)\left\|h_i^t - \nabla f_i(x^t) + \lambda \mathcal{C}_i^t\big(\nabla f_i(x^t) - h_i^t\big)\right\|^2 \\
&\quad + (1+s^{-1})\left\|\nabla f_i(x^{t+1}) - \nabla f_i(x^t)\right\|^2 \\
&\leq (1+s)\left\|h_i^t - \nabla f_i(x^t) + \lambda \mathcal{C}_i^t\big(\nabla f_i(x^t) - h_i^t\big)\right\|^2 \\
&\quad + (1+s^{-1})L_i^2\left\|x^{t+1} - x^t\right\|^2.
\end{aligned}
$$

Moreover, conditionally on $x^t$, $h^t$ and $(h_i^t)_{i=1}^n$,

$$
\begin{aligned}
\mathbb{E}\left[\left\|h_i^t - \nabla f_i(x^t) + \lambda \mathcal{C}_i^t\big(\nabla f_i(x^t) - h_i^t\big)\right\|^2\right] &= \left\|h_i^t - \nabla f_i(x^t) + \lambda \mathbb{E}\big[\mathcal{C}_i^t\big(\nabla f_i(x^t) - h_i^t\big)\big]\right\|^2 \\
&\quad + \lambda^2 \mathbb{E}\left[\left\|\mathcal{C}_i^t\big(\nabla f_i(x^t) - h_i^t\big) - \mathbb{E}\big[\mathcal{C}_i^t\big(\nabla f_i(x^t) - h_i^t\big)\big]\right\|^2\right] \\
&\leq \left\|h_i^t - \nabla f_i(x^t) + \lambda \mathbb{E}\big[\mathcal{C}_i^t\big(\nabla f_i(x^t) - h_i^t\big)\big]\right\|^2 \\
&\quad + \lambda^2 \omega \left\|\nabla f_i(x^t) - h_i^t\right\|^2.
\end{aligned}
$$

In addition,

$$
\begin{aligned}
\left\|h_i^t - \nabla f_i(x^t) + \lambda \mathbb{E}\big[\mathcal{C}_i^t\big(\nabla f_i(x^t) - h_i^t\big)\big]\right\| &\leq \left\|\lambda\big(h_i^t - \nabla f_i(x^t)\big) + \lambda \mathbb{E}\big[\mathcal{C}_i^t\big(\nabla f_i(x^t) - h_i^t\big)\big]\right\| \\
&\quad + (1-\lambda)\left\|h_i^t - \nabla f_i(x^t)\right\| \\
&\leq \lambda \eta \left\|\nabla f_i(x^t) - h_i^t\right\| + (1-\lambda)\left\|\nabla f_i(x^t) - h_i^t\right\| \\
&= (1 - \lambda + \lambda\eta)\left\|\nabla f_i(x^t) - h_i^t\right\|.
\end{aligned}
$$

Therefore, conditionally on $x^t$, $h^t$ and $(h_i^t)_{i=1}^n$,

$$
\mathbb{E}\left[\left\|h_i^t - \nabla f_i(x^t) + \lambda \mathcal{C}_i^t\big(\nabla f_i(x^t) - h_i^t\big)\right\|^2\right] \leq \big((1 - \lambda + \lambda\eta)^2 + \lambda^2 \omega\big)\left\|\nabla f_i(x^t) - h_i^t\right\|^2
$$

and

$$
\begin{aligned}
\mathbb{E}\left[\left\|\nabla f_i(x^{t+1}) - h_i^{t+1}\right\|^2\right] &\leq (1+s)\big((1 - \lambda + \lambda\eta)^2 + \lambda^2 \omega\big)\left\|\nabla f_i(x^t) - h_i^t\right\|^2 \\
&\quad + (1+s^{-1})L_i^2\mathbb{E}\left[\left\|x^{t+1} - x^t\right\|^2\right],
\end{aligned}
$$

so that

$$
\begin{aligned}
\mathbb{E}\left[\frac{1}{n}\sum_{i=1}^n \left\|\nabla f_i(x^{t+1}) - h_i^{t+1}\right\|^2\right] &\leq (1+s)\big((1 - \lambda + \lambda\eta)^2 + \lambda^2 \omega\big)\frac{1}{n}\sum_{i=1}^n \left\|\nabla f_i(x^t) - h_i^t\right\|^2 \\
&\quad + (1+s^{-1})\tilde{L}^2\mathbb{E}\left[\left\|x^{t+1} - x^t\right\|^2\right].
\end{aligned}
$$

Let $\theta > 0$; its value will be set to $\theta^\star$ later on. We introduce the Lyapunov function, for every $t \geq 0$,

$$
\Psi^t := f(x^t) - f^\star + \frac{\gamma}{2\theta}\frac{1}{n}\sum_{i=1}^n \left\|\nabla f_i(x^t) - h_i^t\right\|^2.
$$

Hence, for every $t \geq 0$, conditionally on $x^t$, $h^t$ and $(h_i^t)_{i=1}^n$, we have

$$
\begin{aligned}
\mathbb{E}\big[\Psi^{t+1}\big] &\leq (1 - \gamma\mu)\big(f(x^t) - f^\star\big) \\
&\quad + \frac{\gamma}{2\theta}\Big(\theta\big((1 - \nu + \nu\eta)^2 + \nu^2\omega_{\mathrm{av}}\big) \\
&\quad + (1+s)\big((1 - \lambda + \lambda\eta)^2 + \lambda^2\omega\big)\Big)\frac{1}{n}\sum_{i=1}^n \left\|\nabla f_i(x^t) - h_i^t\right\|^2 \qquad (17) \\
&\quad + \left(\frac{L}{2} - \frac{1}{2\gamma} + \frac{\gamma}{2\theta}(1+s^{-1})\tilde{L}^2\right)\mathbb{E}\left[\left\|x^{t+1} - x^t\right\|^2\right].
\end{aligned}
$$

Making use of $r$ and $r_{\mathrm{av}}$ and setting $\theta = s(1+s)\frac{r}{r_{\mathrm{av}}}$, we can rewrite (17) as:

$$\mathbb{E}\big[\Psi^{t+1}\big] \leq (1 - \gamma\mu)\big(f(x^t) - f^\star\big) + \frac{\gamma}{2\theta}\Big(\theta r_{\mathrm{av}} + (1+s)r\Big)\frac{1}{n}\sum_{i=1}^n \big\|\nabla f_i(x^t) - h_i^t\big\|^2$$

$$+ \left(\frac{L}{2} - \frac{1}{2\gamma} + \frac{\gamma}{2\theta}(1 + s^{-1})\tilde{L}^2\right)\mathbb{E}\big[\big\|x^{t+1} - x^t\big\|^2\big]$$

$$= (1 - \gamma\mu)\big(f(x^t) - f^\star\big) + \frac{\gamma}{2\theta}(1+s)^2\frac{r}{n}\sum_{i=1}^n \big\|\nabla f_i(x^t) - h_i^t\big\|^2$$

$$+ \left(\frac{L}{2} - \frac{1}{2\gamma} + \frac{\gamma}{2s^2}\frac{r_{\mathrm{av}}}{r}\tilde{L}^2\right)\mathbb{E}\big[\big\|x^{t+1} - x^t\big\|^2\big].$$

We now choose $\gamma$ small enough so that

$$L - \frac{1}{\gamma} + \frac{\gamma}{s^2}\frac{r_{\mathrm{av}}}{r}\tilde{L}^2 \leq 0. \tag{18}$$

A sufficient condition for (18) to hold is (Richtárik et al., 2021, Lemma 5):

$$0 < \gamma \leq \frac{1}{L + \tilde{L}\sqrt{\frac{r_{\mathrm{av}}}{r}\frac{1}{s}}}. \tag{19}$$

Then, assuming that (19) holds, we have, for every $t \geq 0$, conditionally on $x^t$, $h^t$ and $(h_i^t)_{i=1}^n$,

$$\mathbb{E}\big[\Psi^{t+1}\big] \leq (1 - \gamma\mu)\big(f(x^t) - f^\star\big) + \frac{\gamma}{2\theta}(1+s)^2\frac{r}{n}\sum_{i=1}^n \big\|\nabla f_i(x^t) - h_i^t\big\|^2$$

$$\leq \max\big(1 - \gamma\mu, (1+s)^2 r\big)\Psi^t.$$

We see that $s$ must be small enough so that $(1 + s)^2 r < 1$; this is the case with $s = s^\star$, so that $(1 + s^\star)^2 r = \frac{r+1}{2} < 1$. Therefore, we set $s = s^\star$, and, accordingly, $\theta = \theta^\star$. Then, for every $t \geq 0$, conditionally on $x^t$, $h^t$ and $(h_i^t)_{i=1}^n$,

$$\mathbb{E}\big[\Psi^{t+1}\big] \leq \max\big(1 - \gamma\mu, \frac{r+1}{2}\big)\Psi^t.$$

Unrolling the recursion using the tower rule yields (9).

## G  Proof of Theorem 2

Using $L$-smoothness of $f$, we have, for every $t \geq 0$,

$$f(x^{t+1}) \leq f(x^t) + \langle \nabla f(x^t), x^{t+1} - x^t \rangle + \frac{L}{2}\|x^{t+1} - x^t\|^2.$$

Moreover, using convexity of $R$, we have, for every subgradient $u^{t+1} \in \partial R(x^{t+1})$,

$$R(x^t) \geq R(x^{t+1}) + \langle u^{t+1}, x^t - x^{t+1} \rangle. \tag{20}$$

From the property that $\mathrm{prox}_{\gamma R} = (\mathrm{Id} + \gamma\partial R)^{-1}$ (Bauschke & Combettes, 2017), it follows from $x^{t+1} = \mathrm{prox}_{\gamma R}(x^t - \gamma g^{t+1})$ that

$$0 \in \partial R(x^{t+1}) + \frac{1}{\gamma}(x^{t+1} - x^t + \gamma g^{t+1}).$$

So, we set $u^{t+1} := \frac{1}{\gamma}(x^t - x^{t+1}) - g^{t+1}$. Using this subgradient in (20) and replacing $x^t - x^{t+1}$ by $\gamma(u^{t+1} + g^{t+1})$, we get, for every $t \geq 0$,

$$
\begin{aligned}
f(x^{t+1}) + R(x^{t+1}) &\leq f(x^t) + R(x^t) + \langle \nabla f(x^t) + u^{t+1}, x^{t+1} - x^t \rangle + \frac{L}{2}\|x^{t+1} - x^t\|^2 \\
&= f(x^t) + R(x^t) - \gamma\langle \nabla f(x^t) + u^{t+1}, g^{t+1} + u^{t+1} \rangle + \frac{L}{2}\gamma^2\|g^{t+1} + u^{t+1}\|^2 \\
&= f(x^t) + R(x^t) + \frac{\gamma}{2}\|\nabla f(x^t) - g^{t+1}\|^2 + \left(\frac{\gamma^2 L}{2} - \frac{\gamma}{2}\right)\|g^{t+1} + u^{t+1}\|^2 \\
&\quad - \frac{\gamma}{2}\|\nabla f(x^t) + u^{t+1}\|^2 \\
&= f(x^t) + R(x^t) + \frac{\gamma}{2}\|\nabla f(x^t) - g^{t+1}\|^2 + \left(\frac{L}{2} - \frac{1}{2\gamma}\right)\|x^{t+1} - x^t\|^2 \\
&\quad - \frac{\gamma}{2}\|\nabla f(x^t) + u^{t+1}\|^2
\end{aligned}
$$

Note that we recover (16) if $R = 0$ and $u^t \equiv 0$.

Using the fact that for any vectors $a$ and $b$, $-\|a + b\|^2 \leq -\frac{1}{2}\|a\|^2 + \|b\|^2$, we have, for every $t \geq 0$,

$$
\begin{aligned}
-\frac{\gamma}{2}\|\nabla f(x^t) + u^{t+1}\|^2 &\leq -\frac{\gamma}{4}\|\nabla f(x^{t+1}) + u^{t+1}\|^2 + \frac{\gamma}{2}\|\nabla f(x^{t+1}) - \nabla f(x^t)\|^2 \\
&\leq -\frac{\gamma}{4}\|\nabla f(x^{t+1}) + u^{t+1}\|^2 + \frac{\gamma L^2}{2}\|x^{t+1} - x^t\|^2.
\end{aligned}
$$

Hence, for every $t \geq 0$,

$$
\begin{aligned}
f(x^{t+1}) + R(x^{t+1}) &\leq f(x^t) + R(x^t) + \frac{\gamma}{2}\|\nabla f(x^t) - g^{t+1}\|^2 + \left(\frac{L}{2} - \frac{1}{2\gamma} + \frac{\gamma L^2}{2}\right)\|x^{t+1} - x^t\|^2 \\
&\quad - \frac{\gamma}{4}\|\nabla f(x^{t+1}) + u^{t+1}\|^2.
\end{aligned}
$$

It follows from the KŁ assumption (7) that

$$
\begin{aligned}
f(x^{t+1}) + R(x^{t+1}) - f^\star - R^\star &\leq f(x^t) + R(x^t) - f^\star - R^\star + \frac{\gamma}{2}\|\nabla f(x^t) - g^{t+1}\|^2 \\
&\quad + \left(\frac{L}{2} - \frac{1}{2\gamma} + \frac{\gamma L^2}{2}\right)\|x^{t+1} - x^t\|^2 \\
&\quad - 2\mu\frac{\gamma}{4}\left(f(x^{t+1}) + R(x^{t+1}) - f^\star - R^\star\right),
\end{aligned}
$$

so that

$$
\begin{aligned}
\left(1 + \frac{\gamma\mu}{2}\right)\left(f(x^{t+1}) + R(x^{t+1}) - f^\star - R^\star\right) &\leq f(x^t) + R(x^t) - f^\star - R^\star + \frac{\gamma}{2}\|\nabla f(x^t) - g^{t+1}\|^2 \\
&\quad + \left(\frac{L}{2} - \frac{1}{2\gamma} + \frac{\gamma L^2}{2}\right)\|x^{t+1} - x^t\|^2,
\end{aligned}
$$

and

$$
\begin{aligned}
f(x^{t+1}) + R(x^{t+1}) - f^\star - R^\star &\leq \left(1 + \frac{\gamma\mu}{2}\right)^{-1}\left(f(x^t) + R(x^t) - f^\star - R^\star\right) + \frac{\gamma}{2}\|\nabla f(x^t) - g^{t+1}\|^2 \\
&\quad + \left(\frac{L}{2} - \frac{1}{2\gamma} + \frac{\gamma L^2}{2}\right)\|x^{t+1} - x^t\|^2.
\end{aligned}
$$

Let $s > 0$. Like in the proof of Theorem 1, we have

$$
\begin{aligned}
\mathbb{E}\left[\frac{1}{n}\sum_{i=1}^n \|\nabla f_i(x^{t+1}) - h_i^{t+1}\|^2\right] &\leq (1+s)\left((1 - \lambda + \lambda\eta)^2 + \lambda^2\omega\right)\frac{1}{n}\sum_{i=1}^n \|\nabla f_i(x^t) - h_i^t\|^2 \\
&\quad + (1 + s^{-1})\tilde{L}^2\mathbb{E}\left[\|x^{t+1} - x^t\|^2\right]
\end{aligned}
$$

and

$$\mathbb{E}\left[\left\|g^{t+1} - \nabla f(x^t)\right\|^2\right] \leq \left((1 - \nu + \nu\eta)^2 + \nu^2\omega_{\mathrm{av}}\right)\frac{1}{n}\sum_{i=1}^n \left\|\nabla f_i(x^t) - h_i^t\right\|^2.$$

We introduce the Lyapunov function, for every $t \geq 0$,

$$\Psi^t := f(x^t) + R(x^t) - f^\star - R^\star + \frac{\gamma}{2\theta}\frac{1}{n}\sum_{i=1}^n \left\|\nabla f_i(x^t) - h_i^t\right\|^2,$$

where $\theta = s(1+s)\frac{r}{r_{\mathrm{av}}}$.

Following the same derivations as in the proof of Theorem 1, we obtain that, for every $t \geq 0$, conditionally on $x^t$, $h^t$ and $(h_i^t)_{i=1}^n$,

$$
\begin{aligned}
\mathbb{E}\left[\Psi^{t+1}\right] &\leq \left(1 + \frac{\gamma\mu}{2}\right)^{-1}\left(f(x^t) + R(x^t) - f^\star - R^\star\right) \\
&\quad + \frac{\gamma}{2\theta}\Big(\theta\big((1 - \nu + \nu\eta)^2 + \nu^2\omega_{\mathrm{av}}\big) \\
&\quad + (1+s)\big((1 - \lambda + \lambda\eta)^2 + \lambda^2\omega\big)\Big)\frac{1}{n}\sum_{i=1}^n \left\|\nabla f_i(x^t) - h_i^t\right\|^2 \\
&\quad + \left(\frac{L}{2} - \frac{1}{2\gamma} + \frac{\gamma L^2}{2} + \frac{\gamma}{2\theta}(1 + s^{-1})\tilde{L}^2\right)\mathbb{E}\left[\left\|x^{t+1} - x^t\right\|^2\right] \\
&= \left(1 + \frac{\gamma\mu}{2}\right)^{-1}\left(f(x^t) + R(x^t) - f^\star - R^\star\right) \\
&\quad + \frac{\gamma}{2\theta}\Big(\theta r_{\mathrm{av}} + (1+s)r\Big)\frac{1}{n}\sum_{i=1}^n \left\|\nabla f_i(x^t) - h_i^t\right\|^2 \\
&\quad + \left(\frac{L}{2} - \frac{1}{2\gamma} + \frac{\gamma L^2}{2} + \frac{\gamma}{2\theta}(1 + s^{-1})\tilde{L}^2\right)\mathbb{E}\left[\left\|x^{t+1} - x^t\right\|^2\right] \\
&= \left(1 + \frac{\gamma\mu}{2}\right)^{-1}\left(f(x^t) + R(x^t) - f^\star - R^\star\right) + \frac{\gamma}{2\theta}(1+s)^2\frac{r}{n}\sum_{i=1}^n \left\|\nabla f_i(x^t) - h_i^t\right\|^2 \\
&\quad + \left(\frac{L}{2} - \frac{1}{2\gamma} + \frac{\gamma L^2}{2} + \frac{\gamma}{2s^2}\frac{r_{\mathrm{av}}}{r}\tilde{L}^2\right)\mathbb{E}\left[\left\|x^{t+1} - x^t\right\|^2\right].
\end{aligned}
$$

We now choose $\gamma$ small enough so that

$$L - \frac{1}{\gamma} + \gamma L^2 + \frac{\gamma}{s^2}\frac{r_{\mathrm{av}}}{r}\tilde{L}^2 \leq 0.$$

If we assume $\gamma \leq \frac{1}{L}$, a sufficient condition is

$$2L - \frac{1}{\gamma} + \frac{\gamma}{s^2}\frac{r_{\mathrm{av}}}{r}\tilde{L}^2 \leq 0. \tag{21}$$

A sufficient condition for (21) to hold is (Richtárik et al., 2021, Lemma 5):

$$0 < \gamma \leq \frac{1}{2L + \tilde{L}\sqrt{\frac{r_{\mathrm{av}}}{r}\frac{1}{s}}}. \tag{22}$$

Then, assuming that (22) holds, we have, for every $t \geq 0$, conditionally on $x^t$, $h^t$ and $(h_i^t)_{i=1}^n$,

$$
\begin{aligned}
\mathbb{E}\left[\Psi^{t+1}\right] &\leq \left(1 + \frac{\gamma\mu}{2}\right)^{-1}\left(f(x^t) + R(x^t) - f^\star - R^\star\right) + \frac{\gamma}{2\theta}(1+s)^2\frac{r}{n}\sum_{i=1}^n \left\|\nabla f_i(x^t) - h_i^t\right\|^2 \\
&\leq \max\left(\frac{1}{1 + \frac{1}{2}\gamma\mu}, (1+s)^2 r\right)\Psi^t.
\end{aligned}
$$

We set $s = s^\star$ and, accordingly, $\theta = \theta^\star$, so that $(1 + s^\star)^2 r = \frac{r+1}{2} < 1$. Then, for every $t \geq 0$, conditionally on $x^t$, $h^t$ and $(h_i^t)_{i=1}^n$,

$$\mathbb{E}\left[\Psi^{t+1}\right] \leq \max\left(\frac{1}{1 + \frac{1}{2}\gamma\mu}, \frac{r+1}{2}\right)\Psi^t.$$

Unrolling the recursion using the tower rule yields (11).

# H  Proof of Theorem 3

Let $\theta > 0$; its value will be set to the prescribed value later on. We introduce the Lyapunov function, for every $t \geq 0$,

$$\Psi^t := f(x^t) - f^{\inf} + \frac{\gamma}{2\theta} \frac{1}{n} \sum_{i=1}^{n} \left\| \nabla f_i(x^t) - h_i^t \right\|^2.$$

According to (Richtárik et al., 2021, Lemma 4), we have, for every $t \geq 0$,

$$f(x^{t+1}) - f^{\inf} \leq f(x^t) - f^{\inf} - \frac{\gamma}{2} \left\| \nabla f(x^t) \right\|^2 + \frac{\gamma}{2} \left\| g^{t+1} - \nabla f(x^t) \right\|^2 + \left( \frac{L}{2} - \frac{1}{2\gamma} \right) \left\| x^{t+1} - x^t \right\|^2.$$

As shown in the proof of Theorem 1, we have, conditionally on $x^t$, $h^t$ and $(h_i^t)_{i=1}^{n}$,

$$\mathbb{E}\left[ \left\| g^{t+1} - \nabla f(x^t) \right\|^2 \right] \leq \left( (1 - \nu + \nu\eta)^2 + \nu^2 \omega_{\mathrm{av}} \right) \frac{1}{n} \sum_{i=1}^{n} \left\| \nabla f_i(x^t) - h_i^t \right\|^2.$$

As for the control variates $h_i^t$, as shown in the proof of Theorem 1, we have, conditionally on $x^t$, $h^t$ and $(h_i^t)_{i=1}^{n}$,

$$\mathbb{E}\left[ \frac{1}{n} \sum_{i=1}^{n} \left\| \nabla f_i(x^{t+1}) - h_i^{t+1} \right\|^2 \right] \leq (1+s)\left( (1 - \lambda + \lambda\eta)^2 + \lambda^2 \omega \right) \frac{1}{n} \sum_{i=1}^{n} \left\| \nabla f_i(x^t) - h_i^t \right\|^2$$
$$+ (1 + s^{-1}) \tilde{L}^2 \mathbb{E}\left[ \left\| x^{t+1} - x^t \right\|^2 \right].$$

Hence, for every $t \geq 0$, conditionally on $x^t$, $h^t$ and $(h_i^t)_{i=1}^{n}$, we have

$$\mathbb{E}\left[ \Psi^{t+1} \right] \leq f(x^t) - f^{\inf} - \frac{\gamma}{2} \left\| \nabla f(x^t) \right\|^2$$
$$+ \frac{\gamma}{2\theta} \left( \theta\left( (1 - \nu + \nu\eta)^2 + \nu^2 \omega_{\mathrm{av}} \right) + (1+s)\left( (1 - \lambda + \lambda\eta)^2 + \lambda^2 \omega \right) \right) \frac{1}{n} \sum_{i=1}^{n} \left\| \nabla f_i(x^t) - h_i^t \right\|^2$$
$$+ \left( \frac{L}{2} - \frac{1}{2\gamma} + \frac{\gamma}{2\theta}(1 + s^{-1}) \tilde{L}^2 \right) \mathbb{E}\left[ \left\| x^{t+1} - x^t \right\|^2 \right]. \tag{23}$$

Let $r := (1 - \lambda + \lambda\eta)^2 + \lambda^2 \omega$, $r_{\mathrm{av}} := (1 - \nu + \nu\eta)^2 + \nu^2 \omega_{\mathrm{av}}$. Set $\theta := s(1+s)\frac{r}{r_{\mathrm{av}}}$. We can rewrite (23) as:

$$\mathbb{E}\left[ \Psi^{t+1} \right] \leq f(x^t) - f^{\inf} - \frac{\gamma}{2} \left\| \nabla f(x^t) \right\|^2 + \frac{\gamma}{2\theta} \left( \theta r_{\mathrm{av}} + (1+s)r \right) \frac{1}{n} \sum_{i=1}^{n} \left\| \nabla f_i(x^t) - h_i^t \right\|^2$$
$$+ \left( \frac{L}{2} - \frac{1}{2\gamma} + \frac{\gamma}{2\theta}(1 + s^{-1}) \tilde{L}^2 \right) \mathbb{E}\left[ \left\| x^{t+1} - x^t \right\|^2 \right]$$
$$= f(x^t) - f^{\inf} - \frac{\gamma}{2} \left\| \nabla f(x^t) \right\|^2 + \frac{\gamma}{2\theta}(1+s)^2 \frac{r}{n} \sum_{i=1}^{n} \left\| \nabla f_i(x^t) - h_i^t \right\|^2$$
$$+ \left( \frac{L}{2} - \frac{1}{2\gamma} + \frac{\gamma}{2s^2} \frac{r_{\mathrm{av}}}{r} \tilde{L}^2 \right) \mathbb{E}\left[ \left\| x^{t+1} - x^t \right\|^2 \right].$$

We now choose $\gamma$ small enough so that

$$L - \frac{1}{\gamma} + \frac{\gamma}{s^2} \frac{r_{\mathrm{av}}}{r} \tilde{L}^2 \leq 0. \tag{24}$$

A sufficient condition for (24) to hold is (Richtárik et al., 2021, Lemma 5):

$$0 < \gamma \leq \frac{1}{L + \tilde{L}\sqrt{\frac{r_{\mathrm{av}}}{r}} \frac{1}{s}}. \tag{25}$$

Then, assuming that (25) holds, we have, for every $t \geq 0$, conditionally on $x^t$, $h^t$ and $(h_i^t)_{i=1}^{n}$,

$$\mathbb{E}\left[ \Psi^{t+1} \right] \leq f(x^t) - f^{\inf} - \frac{\gamma}{2} \left\| \nabla f(x^t) \right\|^2 + \frac{\gamma}{2\theta}(1+s)^2 \frac{r}{n} \sum_{i=1}^{n} \left\| \nabla f_i(x^t) - h_i^t \right\|^2.$$

We have chosen $s$ so that $(1+s)^2 r = 1$. Hence, using the tower rule, we have, for every $t \geq 0$,

$$\mathbb{E}\big[\Psi^{t+1}\big] \leq \mathbb{E}\big[\Psi^t\big] - \frac{\gamma}{2}\mathbb{E}\big[\|\nabla f(x^t)\|^2\big].$$

Let $T \geq 1$. By summing up the inequalities for $t = 0, \cdots, T-1$, we get

$$0 \leq \mathbb{E}\big[\Psi^T\big] \leq \Psi^0 - \frac{\gamma}{2}\sum_{t=0}^{T-1}\mathbb{E}\big[\|\nabla f(x^t)\|^2\big].$$

Multiplying both sides by $\frac{2}{\gamma T}$ and rearranging the terms, we get

$$\frac{1}{T}\sum_{t=0}^{T-1}\mathbb{E}\big[\|\nabla f(x^t)\|^2\big] \leq \frac{2}{\gamma T}\Psi^0,$$

where the left hand side can be interpreted as $\mathbb{E}\big[\|\nabla f(\hat{x}^T)\|^2\big]$, where $\hat{x}^T$ is chosen from $x^0, x^1, \ldots, x^{T-1}$ uniformly at random.