# OpenReview forum: "EF-BV: A Unified Theory of Error Feedback and Variance Reduction Mechanisms for Biased and Unbiased Compression in Distributed Optimization"
_NeurIPS.cc/2022/Conference — NeurIPS 2022 Accept_

### Official Review · Reviewer_6of8 · 2022-06-29

**Rating:** 5
**Confidence:** 4
**Soundness:** 3 good
**Presentation:** 3 good
**Contribution:** 3 good

**Summary:**

This paper introduced a new communication-efficient distributed training method called EF-BV, which can be treated as a combination of previous method EF21 and DIANA.  The paper gives its linear convergence analysis under certain  conditions in the main part and experiments on relatively small dataset is provided in the appendix. This paper is well written  and easy to follow.

**Questions:**

See Weaknesses.

**Limitations:**

The paper discusses a new variant on a technique in distributed training. As far as I’m concerned, there is no serious issue or limitation that would impact society.

**Strengths And Weaknesses:**

Strengths: This paper proposed EF-BV which includes unbiased and biased compressors as its particular cases. The linear convergence analyses are detailed and technically sound. Assumptions underlying the proposed method are simpler compared to previous methods.

Weaknesses:

1.The main part can be more concise (especially for the introduction part)and including empirical results.

2.Given the new introduced hyper-parameters, it is still not clear whether this new proposed method is empirically useful. How to choose  hyper-parameters in a more practical training setting?

3.The empirical evaluations can not well supported their theoretical analysis. As the authors claim running experiments with 24 A100 GPUs, all methods should be compared in a relatively large scaled training task. Only small linear regression experiment results are reported, where communication is not really an issue.

---

> ### Author Response · Authors · 2022-08-01
> **Response to your comments and question**
>
> >The main part can be more concise (especially for the introduction part) and including empirical results.
>
> $\rightarrow$ Upon acceptance, we will make use of an additional page to reorganize the paper and put empirical results in the main part.
>
> >Given the new introduced hyper-parameters, it is still not clear whether this new proposed method is empirically useful. How to choose hyper-parameters in a more practical training setting?
>
> $\rightarrow$ EF-BV has one additional hyper-parameter, which is $\nu$. All else is as in other similar methods involving communication compression (e.g., EF21 and DIANA). As indicated in Remark 1 (line 274), there is no need to tune the parameters, since their values should be set as the optimal values given in line 257. So, the improvement of EF-BV over the state of the art EF21 comes essentially for free.
>
> >Only small linear regression experiment results are reported, where communication is not really an issue.
>
> $\rightarrow$ Yes. We are currently running experiments in various settings, including large-scale training tasks. We will include new results in the final paper. However, please note that our experiments are designed to merely test our theory, and our tests do so successfully, despite the fact that the datasets used are not large enough to warrant gradient compression. This is completely OK since our experiments nevertheless convincingly support our theoretical findings. Ours is a different kind of an experimental setup than what is typically needed in purely empirical work, where one needs to run methods in much more realistic scenarios to paint a more convincing picture. Since we have strong theory supporting all our methods, we do not need to resort to such testing.
>
> We don't see major concerns about our paper in your review, so if you are satisfied with our response and the addition of the nonconvex convergence result, please increase your score.

---

> > ### Comment · Reviewer_6of8 · 2022-08-09
> > **Thanks for providing the response.**
> >
> > Thanks to the authors for providing the response! I have no more questions left and hope to the empirical results in large-scale training settings.
> >
> > At this point I would like to discuss with fellow reviewers how they think whether the results in current manuscript is solid enough to be accepted. I’ll give 4->5 (can change it in discussion with fellow reviewers) and am ready to move on to a discussion with fellow reviewers.

---

> ### Author Response · Authors · 2022-08-07
> **Reviewer 6of8: What do you think about our rebuttal?**
>
> Dear Reviewer 6of8,
>
> We believe we addressed your concerns. Additionally, we have included an analysis in the smooth nonconvex case; see our post "To all reviewers: new nonconvex analysis" and the revised paper.
>
> We believe your current score does not reflect the import of the theoretical contributions of our work.
>
> We would be delighted to learn what you think!
>
> Kind regards,
>
> authors

---

### Official Review · Reviewer_LBgw · 2022-07-08

**Rating:** 7
**Confidence:** 3
**Soundness:** 3 good
**Presentation:** 3 good
**Contribution:** 3 good

**Summary:**

This work addresses the problem of compressing the information exchanged between the workers and the parameter-server for distributed optimization. There have been several works in the recent literature that consider this, and subsequently propose various compression mechanisms (eg. top-K, rand-K, quantization, etc.) to reduce the communication demand for various iterative algorithms.

In this work, firstly, the authors propose a slightly different perspective towards the characterization of such compressors. Existing works characterize a compressor $C(\cdot)$ using the *expected compression error* defined as $\mathbb{E}\lVert C(x) - x \rVert^2$, where $x \in \mathbb{R}^d$ is the compressor input and $d$ is the dimension. Consequently, the *expected compression error* is upper bounded in different ways, depending on whether the compressor is *unbiased* or *biased but contractive*. In this work, the authors propose an alternative and finer characterization of any compressor by decomposing the *expected compression error* into two terms: The **bias** of the compressor, and the **variance** of the compressor, as follows:

$\mathbb{E}\lVert C(x) - x \rVert^2 = \lVert \mathbb{E}[C(x)] - x \rVert^2+ \mathbb{E}[\lVert C(x) - \mathbb{E}[C(x)] \rVert^2]$,

where the first and second terms on the R.H.S. are the **bias** and **variance** respectively. Consequently, they propose a new class of compressors, $C(\eta, \omega)$, where $\eta$ characterizes the bias, whereas $\omega$ characterizes the variance of the compressor. They show that for appropriate values of $\eta$ and $\omega$, the class $C(\eta, \omega)$ boils down to known classes of compressors, such as *unbiased compressors* or *biased contractive compressors*, implying that $C(\eta, \omega)$ is a more general characterization of compressors.

Following this, they propose the notion of **average relative variance** that characterizes the variance when compressed outputs from different workers are averaged at the parameter server. Furthermore, they also show that the bias and variance of any compressor $C(\cdot) \in C(\eta, \omega)$ can be traded-off by appropriately scaling the output of $C(\cdot)$, i.e. using $\lambda C(\cdot)$ instead of $C(\cdot)$, for some $\lambda \in (0,1]$. They subsequently show that with an appropriate choice of the scaling $\lambda$, the *scaled compressor*, $\lambda C$ can be made contractive with minimal bias (which is desirable).

After this, the authors use this alternative characterization to propose a unified framework to generalize two existing works: **DIANA** *(Mishchenko et al.)* and **EF21** *(Richtarik et al.)*. DIANA studies the utility of unbiased compressors for distributed optimization. At every iteration, DIANA maintains a control variate **h** and predicts the (approximate) gradient from the control variate by doing a first-order predictive coding from compressed gradient differences. To do this prediction, it utilizes a parameter $\lambda$ that controls the variance of the descent direction -- this is also referred to as **Variance Reduction**. On the other hand, EF21 studies biased contractive compressors, and shows that the idea of predictive first-order coding of DIANA can also be extended to mitigate the bias of contractive compressors, as long as the control variates converges to a point, implying that the sequence being compressed goes to 0. EF21 shows that compressing gradient differences (or, first-order predictive coding) implicitly does error-feedback to ensure convergence. However, it does not explicitly do anything to control the variance of the descent steps.

In the context of both of these works, EF-BV (short for **Error-Feedback with Bias-Variance Decomposition**) uses the *bias-variance decomposition* of the class $C(\eta, \omega)$ of compressors, that uses appropriate scaling to do both -- $\rm (i)$ Updating the control variate from compressed gradient difference, and $\rm (ii)$ Obtaining the descent direction from the control variate. To do $\rm (i)$, the scaling is chosen so that the individual compression error at each worker is small, whereas to do to $\rm (ii)$, the scaling is chosen so that the error in the (globally averaged) descent direction at the parameter server is small (this is where the characterization of **average relative variance** comes into picture). To do $\rm (i)$ and $\rm (ii)$, the optimal scaling makes use of the bias-variance decomposition parameters $\eta$ and $\omega$. EF-BV does separate scaling for both, allowing the algorithm designer a finer control on the compression error.

Finally, the authors show how this approach subsumes DIANA and EF-21, while giving analytic convergence results. They prove linear convergence for objective functions that satisfy the Kurdyka-Lojasiewicz (KL) condition, which is a generalization of Polyak-Lojasiewicz (PL) condition for regularized objective functions. To do this, they choose appropriate values of the scaling parameters for $\rm (i)$ and $\rm (ii)$ as mentioned above. Subsequently, they also show how their approach has new implications for DIANA and EF-21.





**Questions:**

The paper is well-written. However, I have some questions and suggestions and I would be grateful if the authors address them. I would be happy to change my rating if these concerns are adequately addressed:

1. The authors propose a general class of compressors $C(\eta, \omega)$, and mention in line 103: "... It deals with unbiased compressors, biased contractive compressors, and possibly even more ..." When I read this line, "possibly even more" made me wonder if it's really that obvious? Does the class $C(\eta, \omega)$ actually include compressors that are not obtained from a simple re-scaling of existing compressors in $\mathbb{B}(\alpha)$ or $\mathbb{U}(\omega)$? The authors do mention **mix-(k,k')** and **comp-(k,k')** -- however, they also note that similar compression errors can be obtained for appropriately designed **top-K** or **rand-K** compressors too. Is it not clear why one would prefer using **mix-(k,k')** or **comp-(k,k')** over these. In other words, the answer of the following question is not obvious to me, "Are there compressors in the new proposed class $C(\eta, \omega)$ that perform better than existing compressors in any sense?" A possible answer to this question might be: Even if the total compression error is the same, does the distinction of bias and variance for these compressors enable EF-BV to achieve a better convergence rate than with simply **top-(k + k')** or **scaled rand-(k + k')**? If what I mention here holds true, the authors should explicitly discuss this somewhere in the paper. Another possible answer to this question might be the concatenated use of compression and quantization operators that provide a different way of trading-off the bias and variance (some relevant references are mentioned in pts. 5-(b) and 5-(c) below).

2. The authors define the **relative average variance** in order to characterize the variance of the compressed outputs averaged at the parameter server. But they make no mention of the **relative average bias** -- is it because for independent and identical compressors at the workers, global averaging at the parameter server makes no difference to the bias? If that is the case, I would request the authors to explicitly mention it somewhere (perhaps a footnote). If average variance is considered, it is only natural to discuss about the average bias, since a primary hinging idea of this work is the bias-variance decomposition of the compressors.

3. The authors should reference the proofs for Propositions 4 and 5 in Appendices. A.1 and A.2, for the bias and variance of **mix-(k,k')** and **comp-(k,k')** to relevant sections App. D and App. E.

4. Since the authors emphasize that the **average relative variance** can be *much smaller* than the individual variances of compressors at the workers due to large number of workers (n), they are requested to add an explicit discussion regarding how the communication / iteration complexity of EF-BV varies with the number of workers (n). Is the improvement only because of appearance of $\omega_{av}$ in place of $\omega$ or are there other expressions too through which "n" affects the convergence rate?

5. The authors have missed several relevant references:

      (a) The authors claim in line 508 that they *"propose"* **comp-(k,k')**: However, a concatenated application of top-k and rand-k has already been studied in the following: ***Leighton Pate Barnes, Huseyin A. Inan, Berivan Isik, Ayfer Ozgur, "rTop-k: A Statistical Estimation Approach to Distributed SGD", https://arxiv.org/abs/2005.10761***.

      (b) More recent works using error-feedback wherein the convergence rate (at least for smooth and strongly convex functions) go beyond just *linear convergence* to *linear convergence with rates approaching uncompressed gradient descent*. In particular, ***Chung-Yi Lin, Victoria Kostina, Babak Hassibi, "Differentially Quantized Gradient Methods", http://128.84.4.18/abs/2002.02508*** studies quantization methods, and uses error feedback to ensure that gradients are *always evaluated* on the unquantized trajectory. Another work: ***Rajarshi Saha, Mert Pilanci, Andrea J. Goldsmith, "Democratic Source Coding: An Optimal Fixed-Length Quantization Scheme for Distributed Optimization Under Communication Constraints", https://arxiv.org/abs/2103.07578*** builds on top of this strategy to achieve dimension independent linear convergence rates close to the unquantized linear convergence rates.

      (c) Some other recent relevant works on quantization for distributed optimization: ***Venkata Gandikota, Daniel Kane, Raj Kumar Maity, Arya Mazumdar, "vqSGD: Vector Quantized Stochastic Gradient Descent", https://arxiv.org/abs/1911.07971*** and ***Prathamesh Mayekar, Himanshu Tyagi, "RATQ: A Universal Fixed-Length Quantizer for Stochastic Optimization", https://arxiv.org/abs/1908.08200*** -- It is important to mention them because quantization is inevitable in the system implementation of any distributed optimization algorithm. So, the bias-variance tradeoffs for concatenated application of various compression and quantization strategies (in order to show that they lie in $C(\eta, \omega)$) is relevant to further justify and substantiate the importance of the new general class of compressors $C(\eta, \omega)$.


**Limitations:**

The work does not have any major limitations or potential negative impact. Some minor suggestions for clarification/improvement have been mentioned in the previous sections **Questions** and **Limitations** of the review.

One thing that I would appreciate if the authors cleared it up for me, "How extensively does the framework proposed in this paper require the independence of the compressors across different workers?" I understand it is not necessary as the authors mention *m-nice sampling* -- however, even for this scheme, the set $\Omega$ of participating workers is selected uniformly at random which simplifies the analysis. Do the authors have any comments on if $\omega_{av}$ is a sufficient way to characterize the average relative variance, and how to derive expressions for $\omega_{av}$ in settings where the compressors across different workers are different? For example, different number of bits used to quantize the updates of each worker? In other words, "Is $\omega_{av}$ still much smaller than $\omega$ in such under a worst-case scenario where the individual compressors at workers are heterogeneous?" *Clarification -- This is not a major limitation of the work, just a suggestion to consider since it is a more practical scenario.* The authors should add a brief discussion on it because communication compression is a relevant aspect of federated learning and the assumptions of i.i.d. compressors or i.i.d. client selection rarely hold in such settings. Discussing this will perhaps broadens the impact of the paper.

Another suggestion I would have for the authors is to have some numerical experiments in the main body of the paper. An application-oriented theoretical research like this work is better appreciated with substantive numerical experiments, especially since the analyses solely can often not be tight. I understand the page limitations, but just a suggestion to put the experiments in the main body of the paper instead of the supplementary, in case the work is accepted and they get an additional page.

**Strengths And Weaknesses:**

**Originality**: The work is mostly original, although it builds heavily on the works of Mishchenko et al. (DIANA) and Richtarik et al. (EF21). The authors have placed their work appropriately in the context of both these works and also discussed implications of their proposed framework for these existing works. The idea of characterizing estimators using the bias-variance decomposition is a common trick in Estimation & Detection Theory and Signal Processing. However, their application in the context of distributed optimization and its implications in tuning hyper-parameters as done in this work is non-trivial.

**Quality**: The quality of this paper is good. The authors have put in appreciable effort in placing their work in the context of existing works and spent a considerable portion of the paper discussing prior works and how this work improves on top of them. The bibliography is mostly adequate -- however they have missed some recent relevant references that I am aware of and have mentioned in pts. 5-(a), 5-(b), and 5-(c) of **Questions** in the next section of the review.

**Clarity**: For a major portion of the paper, the clarity is good and everything is well-explained. It is well-placed in the context of two of the most related works -- DIANA and EF21, and heavily builds on top of them. However, there were some places that confused me a little when I was reading the paper. Some of them might have gotten clear later in the paper (or over multiple reads, i.e. I had to read a couple of paragraphs multiple times to grasp its significance). I have mentioned them as follows:

1. In line 91, the authors write "... DIANA with independent random compressors has a $\frac{1}{n}$ factor in the increase of iteration complexity due to the use of compression. That is, when n is very large, as is typically the case in practice, the iteration complexity does not suffer from the use of compression. EF21 does not have this property: its convergence rate does not depend on the number n of workers." -- It is not clear how DIANA and EF21 are being compared here. The authors say that in both of them, the iteration complexity does not suffer from increasing number of workers (n) -- why the use of "however" then? Are the authors saying -- DIANA performs better if there are more workers, i.e. as 'n' increases, whereas EF21 does not? If that is the case, perhaps this paragraph needs some rephrasing.

2.   In lines 101-106, the authors make some claims regarding the desiderata of a distributed optimization algorithm. under communication constraints. A suggestion -- The authors should explicitly specify how EF-BV addresses them by reverting back to these points later in the paper after EF-BV is introduced.

**Significance**: The contributions of this work are significant. Specifically, I imagine the bias-variance characterization of compressors to have an impact on guiding the design of distributed algorithms for other settings, beyond EF-BV such as fully decentralized or peer-to-peer settings or federated learning with local averaging. In particular, the bias-variance can be traded-off differently beyond just a simple scaling as done in this work.

---

> ### Author Response · Authors · 2022-08-01
> **Response to weaknesses and limitations**
>
> Thank you for your thorough analysis of our work.
>
> > The work is mostly original, although it builds heavily on the works of Mishchenko et al. (DIANA) and Richtarik et al. (EF21)
>
> $\rightarrow$ All innovation stands on the shoulders of giants! Joke aside, DIANA and EF21 are recent and remarkable algorithms; still, we improve upon them. This is significant, because DIANA and EF21 rely on different analyses and proof techniques. In particular, the existing analysis of DIANA recalled in Proposition 3 is different from our new analysis based on the decay of the objective function. Thus, our work is far from being incremental, and it fills a gap between 2 great approaches, building an even more powerful unified framework.
>
> > Are the authors saying: DIANA performs better if there are more workers, i.e. as $n$ increases, whereas EF21 does not? If that is the case, perhaps this paragraph needs some rephrasing.
>
> $\rightarrow$ Yes, this is what me mean: the convergence rate of DIANA improves with more workers (the independent random errors cancel out when averaged), whereas the rate of EF21 is the same, whatever $n$. We will rephrase the sentence.
>
> > A suggestion: The authors should explicitly specify how EF-BV addresses [these desiderata] by reverting back to these points later in the paper after EF-BV is introduced.
>
> $\rightarrow$ We will make use of an additional page in the final version to synthesize how we addressed the different challenges, for instance in a conclusion.
>
> > How extensively does the framework proposed in this paper require the independence of the compressors across different workers? Do the authors have any comments on if omega_av is a sufficient way to characterize the average relative variance?
>
> $\rightarrow$ Yes, we think that $\omega_{\mathrm{av}}$ is the right way to characterize the variance of the compression error after aggregation. EF-BV outperforms EF21 only if $\omega_{\mathrm{av}}$ is smaller than $\omega$. Otherwise, EF-BV just reverts to EF21. That is why we emphasize the independent case and the sampling case, where $\omega_{\mathrm{av}} \approx \omega/n$. There are other settings where this holds, see for instance the permutation compressors, which are not independent, in Szlendak et al. "Permutation compressors for provably faster distributed nonconvex optimization". So, independence is not a requirement.
>
> > How to derive expressions for $\omega_{\mathrm{av}}$ in settings where the compressors across different workers are different?
>
> $\rightarrow$ We assumed the same value of $\omega$ for all workers, but it is indeed of practical interest to consider different compression factors: the communication cost might be low for one worker and high for another one. Also, in the sampling case, the probability of activation could be different for each worker. It is rather easy to extend our derivations to this more general setting. But the paper is already dense, so we preferred keeping the notations and derivations as simple as possible, with one single value $\omega$. For instance, if the compressors are independent and each worker has a variance $\omega_i$, then $\omega_{\mathrm{av}} = (1/n^2)\sum_i \omega_i$ (which reverts to $\omega/n$ if all $\omega_i$ = the same $\omega$). More generally, $\omega_{\mathrm{av}}$ will characterize the average, and not worst-case, behavior, if the workers are heterogenous.
>
> > Another suggestion I would have for the authors is to have some numerical experiments in the main body of the paper.
>
> $\rightarrow$ We agree that it is better to have the experiments in the main paper. Upon acceptance, we will make use of an additional page to reorganize the content and put experiments in the main part.

---

> ### Author Response · Authors · 2022-08-01
> **Response to questions**
>
> > 1. Does the class actually include compressors that are not obtained from a simple re-scaling of existing compressors in $B(\alpha)$ or $U(\omega)$?
>
> $\rightarrow$ Our mix-(k,k') and comp-(k,k') are rather simple examples, but they are not obtained by scaling. comp-(k,k') sends $k$ floats, so it has the same compression factor as top-k and (scaled) rand-k. And indeed, the distinction of bias and variance enables EF-BV with comp-(k,k') to have a better convergence rate than EF21 with top-k and scaled rand-k, because $\gamma$ can be chosen larger, see eq. (8) and (10). This is because $r_{\mathrm{av}}/r < 1$, which is implied by $\omega_{\mathrm{av}} < \omega$. This central property comes from the independent randomness and the fact that we exploit it in the variance part of the bias-variance decomposition. Our new class $C(\eta,\omega)$ opens the door to non-trivial designs. For instance, when the variables are matrices and not vectors, one can play with low-rank decompositions, sketching operators, exploiting various types of sparsity patterns on the rows and columns, block-quantization, and so on.
>
> > 2. [the authors] make no mention of the relative average bias.
>
> $\rightarrow$ The relative average variance is introduced because when averaging independent random variables, the variance decreases, and this is the cornerstone of every randomized algorithm. On the other hand, the `average bias' is just the average of the biases. Since we assume the same $\omega$ and $\eta$ for all workers, the bias after averaging is simply $\eta$. In other words, the variance and bias behave differently after averaging, and this is why the bias-variance decomposition is so important in understanding biased and random processes.
>
> > 3. The authors should reference the proofs for Propositions 4 and 5 in Appendices A.1 and A.2.
>
> $\rightarrow$ Yes, we will add references in the main paper to all parts in the Appendices.
>
> > 4. Is the improvement [of EF-BV] only because of appearance of $\omega_{\mathrm{av}}$ in place of $\omega$ or are there other expressions too through which $n$ affects the convergence rate?
>
> $\rightarrow$ The improvement of EF-BV over EF21 is indeed completely due to the smaller value $\omega_{\mathrm{av}}$ instead of $\omega$ in the expressions of the parameters $\nu$ and $\gamma$ (the larger, the better). If $\omega_{\mathrm{av}} = \omega$ (for instance rand-k with the same randomness is used at every worker (a bad idea)), EF-BV = EF21.
>
> > 5. a. comp-(k,k') has already been studied in Barnes et al.
>
> $\rightarrow$ You are right, we didn't know this paper. We will correct our statements and cite it. In this paper, the interest of comp-(k,k') is shown in a statistical sense and its superiority in practice is shown in several experiments. So, this paper supports the fact that comp-(k,k') has good properties and we should use it. In this paper, comp-(k,k') is viewed as a contractive compressor, like top-k. We go further with our bias-variance decomposition in exploiting its random nature with a larger stepsize $\gamma$ in EF-BV.
>
> > 5. b. Papers Lin et al. and Saha et al.
>
> $\rightarrow$ Thank you for pointing out these very interesting papers, providing information-theoretic insights on the choice of compressors. From a quick look, it seems that the analysis is for the single-worker case, but we will read them carefully to understand to which extent the results are relevant to our setting. In any case, we will cite these nice papers.
>
> > 5. c. Some other recent relevant works on quantization for distributed optimization...
>
> $\rightarrow$ We agree that quantization is inevitable, and for instance top-k or rand-k are followed in practice by some quantization process, since it is clearly a waste to send 32 bits for each of the k floats. For complicated combinations, calculating the parameters $\eta$, $\omega$, $\omega_{\mathrm{av}}$ can be difficult but should be manageable. We will cite the mentioned papers.

---

> ### Author Response · Authors · 2022-08-07
> **Reviewer LBgw: Did we address your concerns?**
>
> Dear Reviewer LBgw,
>
> We believe we addressed your concerns in two posts called "Response to weaknesses and limitations" and "Response to questions". Additionally, we have included an analysis in the smooth nonconvex case; see our post "To all reviewers: new nonconvex analysis" and the revised paper.
>
> We would be delighted to learn what you think!
>
> Kind regards,
>
> authors

---

> > ### Comment · Reviewer_LBgw · 2022-08-09
> > **Acknowledgement of the rebuttal**
> >
> > Dear authors,
> >
> > I thank you for the detailed point-by-point response, and also apologize for not actively participating in the author-reviewer discussions, although I have been staying on track with the rebuttal.
> >
> > All my queries were adequately answered. I trust that the authors will incorporate the additional discussions and limitations as they mentioned in the rebuttal. I believe the characterization of the bias and variance of compressors used in distributed optimization as done in this paper is quite fundamental and warrants acceptance. I still stand by my review and will be increasing my score after the discussion with other reviewers.
> >
> > I thank the authors and sincerely apologize once again for the late response.

---

> > > ### Author Response · Authors · 2022-08-09
> > > **Re: Acknowledgement of the rebuttal**
> > >
> > > Dear Reviewer LBgw,
> > >
> > > > I thank you for the detailed point-by-point response, and also apologize for not actively participating in the author-reviewer discussions, although I have been staying on track with the rebuttal.
> > >
> > > Thanks, appreciated!
> > >
> > > > All my queries were adequately answered.
> > >
> > > We are happy to hear that.
> > >
> > > > I trust that the authors will incorporate the additional discussions and limitations as they mentioned in the rebuttal.
> > >
> > > Indeed, we shall do so.
> > >
> > > > I believe the characterization of the bias and variance of compressors used in distributed optimization as done in this paper is quite fundamental and warrants acceptance. I still stand by my review and will be increasing my score after the discussion with other reviewers.
> > >
> > > Thank you!!
> > >
> > > > I thank the authors and sincerely apologize once again for the late response.
> > >
> > > Thanks for taking some time our your busy schedule. We understand you are volunteering your time - thanks for that. Plus, your insights are helpful to us.

---

### Official Review · Reviewer_HvEY · 2022-07-11

**Rating:** 5
**Confidence:** 5
**Soundness:** 3 good
**Presentation:** 3 good
**Contribution:** 3 good

**Summary:**

This paper unifies the two distinct compress techniques: DIANA and EF21, into a single framework. This new algorithm, which is named EF-BV, has advantages over the combined two techniques. First, EF-BV allows biased compression, which is not allowed by DIANA. Second, EF-BV introduces randomness, which improves the performance compared to EF21. EF-BV's convergence is provided.

**Questions:**

+ The data in the distributed nodes are homogeneous because they are split after the random shuffle. It would be interesting to see the results for heterogeneous as well.

**Limitations:**

+ The result is only for functions satisfying the PL or KL conditions. Though these conditions are weaker than strongly convex, many problems do not satisfy these conditions.

**Strengths And Weaknesses:**

Strengths
+ The new framework allows more compression techniques than previous algorithms.
+ The convergence results improve the existing one for EF21.

Weakness
- Removing the biasedness for DIANA has been done in recent work: Zhang, J., You, K., & Xie, L. (2021). Innovation compression for communication-efficient distributed optimization with linear convergence. _arXiv preprint arXiv:2105.06697_. This result is for decentralized optimization, which is more difficult than the centralized one considered in this paper.
- EF-BV is only compared with EF21 for the biased compression. It would be interesting to see the comparison with DIANA as well.
- Even the benefit of EF-BV over EF21 is not too much in the numerical experiment.

---

> ### Author Response · Authors · 2022-08-01
> **Response to weaknesses and limitations (1)**
>
> >Removing the biasedness for DIANA has been done in recent work: Zhang, J., You, K., & Xie, L. (2021). This result is for decentralized optimization, which is more difficult than the centralized one considered in this paper.
>
> $\rightarrow$ Thank you for pointing out this paper, which is relevant to the topic of distributed optimization with compression. We will cite it in the final version.
> We are familiar with decentralized optimization and there are some significant differences with centralized optimization, in which we can have a regularizer R and communication is asymmetric (there exists versions of EF21 and DIANA with additional model compression from the master to the nodes, and this is certainly possible for EF-BV), and aggregation/averaging of all local vectors is done at every iteration. So, results in the decentralized setting do not apply straightforwardly to the centralized setting, and the two settings are equally difficult with their own features, in our opinion.
> Furthermore, the algorithm presented by Kovalev et al. in "A linearly convergent algorithm for decentralized optimization: Sending less bits for free!", AISTATS 2021, shares with DIANA the property that compression is applied to the difference, or innovation, between a vector (like a gradient) and its estimate by a control variate, but we cannot call it DIANA (DIANA reverts to GD if no compression is applied, but GD cannot be applied in decentralized optimization, since it would require averaging all gradients at every iteration).
> So, it is incorrect to refer to the results in Zhang et al. as related to DIANA. Therefore, the results of Zhang et al., which are valuable, do not remove any merit or novelty to our work; in other words, we don't see any weakness here.
>
> >EF-BV is only compared with EF21 for the biased compression. It would be interesting to see the comparison with DIANA as well.
>
> $\rightarrow$ DIANA has been proposed and is known to converge with unbiased compressors only. Running DIANA with a biased compressor would make this a heuristic, and we would not know how to set the various parameters (e.g., the stepsize). Also, this is not an approach proposed in previous literature, so it not at all clear why this baseline would make sense. Instead, we chose to only compare prior methods supported with theoretical guarantees, and in the regime where these guarantees apply. So, we disagree that comparing with DIANA in the biased compressor case makes sense, or that this would make the paper better. We can do this if this satisfies the reviewer, this is easy to do. But please be aware of the issues we painted above. Please let us know! Also note that with unbiased compressors, DIANA and EF-BV are essentially the same, as indicated in line 326. Our contribution is the new algorithm EF-BV, which can be used with a large class of possibly biased compressors. Therefore, we put the emphasis on the interest of EF-BV with biased compressors. We do prove convergence of DIANA with biased compressors in Section 4.2 (lines 320-321), but this is just because we call DIANA the particular case of EF-BV with nu=1, which remains new in that case. Therefore, we illustrate the advantage of EF-BV over the state-of-the-art algorithm with biased compressors, which is EF21.
>
> >The benefit of EF-BV over EF21 is not much in the numerical experiments.
>
> $\rightarrow$ Yes. This is because EF21 is already a very good algorithm (indeed, the current theoretical and practical SOTA for error feedback methods handling biased compressors). Please note that the EF21 paper was an oral paper at a recent NeurIPS conference (less than 1% acceptance rate) - it is therefore remarkable that we are able to further improve upon it, and we do so by simply changing the value of one real parameter, $\nu$, from $\nu=\lambda$ to the better choice $\nu=\nu^\star$. This, were believe, is very elegant and beautiful. So, the improvement of EF-BV over EF21 is essentially free and should be known by the community. It is made possible by our detailed analysis and exploitation of the bias and variance properties of compressors.
>
> > The data in the distributed nodes are homogeneous because they are split after the random shuffle. It would be interesting to see the results for heterogeneous as well.
>
> $\rightarrow$ We shuffle the data to be independent from the initial ordering of the samples and to assign the data pieces to the nodes in a fair way. If the dataset is heterogeneous, i.e. there is variability in the samples, there remains variability after shuffling and splitting, with high probability. But we agree that it will be good to run experiments where the heterogeneity is `guaranteed'. We will do so.

---

> ### Author Response · Authors · 2022-08-01
> **Response to weaknesses and limitations (2)**
>
> > The result is only for functions satisfying the PL or KL conditions. Though these conditions are weaker than strongly convex, many problems do not satisfy them.
>
> $\rightarrow$ Obtaining linear convergence guarantees with biased compressors is already an important achievement: before 2021 and the introduction of EF21, no linearly-convergent algorithm was known. We agree that it is even better to have results in other settings. We are working on this and we already derived a result in the nonconvex setting, which we have added in Appendix H. So, if you are satisfied with this addition, please increase your score.

---

> ### Author Response · Authors · 2022-08-07
> **Reviewer HvEY: What do you think about our Author Response?**
>
> Reviewer HvEY,
>
> We believe we addressed your concerns in two posts called "Response to weaknesses and limitations (1)" and "Response to weaknesses and limitations (2)". Additionally, we have included an analysis in the smooth nonconvex case; see our post "To all reviewers: new nonconvex analysis" and the revised paper.
>
> We would be delighted to learn what you think!
>
> Kind regards,
>
> authors

---

### Author Response · Authors · 2022-08-01
**To all reviewers: new nonconvex analysis**

We have **added** at the end of the paper, in Appendix H, a new convergence result in the **nonconvex case**. In the final version, the result will be integrated in the main part of the paper, but in the time being, we keep it separate for the reviewers' convenience. This addition answers the limitation raised by some, that only the PL/KL case was considered.

---

### Meta-Review · Area_Chair_wnTU · 2022-08-25

**Recommendation:** Accept
**Confidence:** Less certain

**Metareview:**

It is strongly suggested by the reviewers that the authors explicitly mention the limitations in their paper -- essentially everything that came out of the discussion period, and not exaggerate the results. It has been noted that the advantage of EF BV over existing EF21 scheme is marginal. Further, authors seem to be oblivious of some relevant works on gradient quantization. However, the generalization framework of error-feedback is an interesting contribution and the community will be benefited from this knowledge.

**Award:**

No

---

### Decision · Program_Chairs · 2022-09-14

Accept